# Metabolic constraints on the evolution of antibiotic resistance

Mattia Zampieri[1,*], Tim Enke[1,2], Victor Chubukov[1], Vito Ricci[3], Laura Piddock[3] & Uwe Sauer[1,**] iD

## Abstract

Despite our continuous improvement in understanding antibiotic resistance, the interplay between natural selection of resistance mutations and the environment remains unclear. To investigate the role of bacterial metabolism in constraining the evolution of antibiotic resistance, we evolved *Escherichia coli* growing on glycolytic or gluconeogenic carbon sources to the selective pressure of three different antibiotics. Profiling more than 500 intracellular and extracellular putative metabolites in 190 evolved populations revealed that carbon and energy metabolism strongly constrained the evolutionary trajectories, both in terms of speed and mode of resistance acquisition. To interpret and explore the space of metabolome changes, we developed a novel constraint-based modeling approach using the concept of shadow prices. This analysis, together with genome resequencing of resistant populations, identified condition-dependent compensatory mechanisms of antibiotic resistance, such as the shift from respiratory to fermentative metabolism of glucose upon overexpression of efflux pumps. Moreover, metabolome-based predictions revealed emerging weaknesses in resistant strains, such as the hypersensitivity to fosfomycin of ampicillin-resistant strains. Overall, resolving metabolic adaptation throughout antibiotic-driven evolutionary trajectories opens new perspectives in the fight against emerging antibiotic resistance.

**Keywords** antibiotic resistance; constraint-based modeling; efflux pump; evolution; metabolism

**Subject Categories** Genome-Scale & Integrative Biology; Metabolism; Microbiology, Virology & Host Pathogen Interaction

**Mol Syst Biol. (2017) 13: 917**

## Introduction

Rapid emergence of multidrug-resistant bacteria renders treatment of bacterial infections once more an urgent global challenge.

Acquired through horizontal gene transfer or genetic mutations, the most effective antibiotic resistance mechanisms alter the antibiotic target, increase drug efflux, or overexpress drug modification enzymes (Blair *et al*, 2015b). While the cost of resistance is highly variable, such resistance mutations or genes often come with a fitness cost that reduces the rate of bacterial proliferation (Dahlberg & Chao, 2003; Melnyk *et al*, 2015). Multiple causes may contribute to reduced fitness, including increased energy and resource demands or activation of less efficient mechanisms that bypass the drug target. The success of resistant mutants critically depends on rapid counterbalancing of the decreased fitness by acquiring compensatory mutations (Levin *et al*, 1997; Marciano *et al*, 2007), which in most cases restore normal growth while preserving resistance to the antibiotics (Marcusson *et al*, 2009). The number and variety of compensatory mutations required to successfully compensate fitness cost varies with organism (Palmer & Kishony, 2013, 2014; Cheng *et al*, 2014) and the particular environmental conditions under which compensation occurs (Testerman *et al*, 2006; Hoffman *et al*, 2010; Toprak *et al*, 2012; Lindsey *et al*, 2013). Nevertheless, the nature of this interaction is poorly understood, and very little is known about the functional constraints that the environment imposes on the evolution of antibiotic resistance and compensatory mechanisms (Björkman *et al*, 2000; King *et al*, 2006; Hoffman *et al*, 2010; Auriol *et al*, 2011; Zhang *et al*, 2011; Villagra *et al*, 2012).

To this end, we investigated metabolic rearrangements during evolution of antibiotic resistance in *Escherichia coli* under two different nutritional conditions. To interpret and understand the impact of metabolic changes in conferring or compensating for antibiotic resistance, we used a genome-scale model of *E. coli* metabolism and developed a novel constraint-based modeling approach. By systematically exploring the space of dual solutions to the linear optimization of flux in each individual reaction, the new approach relates changes of metabolite abundances to potential functional flux rearrangements. This novel systematic approach, together with genome sequence analysis of evolved populations, demonstrates how environmental nutrient composition can directly affect the selection of resistance mechanisms and compensatory mutations.

1 Institute of Molecular Systems Biology, ETH Zürich, Zürich, Switzerland
2 Institute of Biogeochemistry and Pollutant Dynamics (IBP), ETH Zürich, Zürich, Switzerland
3 Institute of Microbiology and Infection, University of Birmingham, Birmingham, UK
*Corresponding author. Tel: +41 44 633 36 89; E-mail: zampieri@imsb.biol.ethz.ch
**Corresponding author. Tel: +41 44 633 36 72; E-mail: sauer@imsb.biol.ethz.ch

# Results

## Generation and metabolic profiling of antibiotic-resistant mutants

To investigate the interplay between evolution of antibiotic resistance and bacterial metabolism, we selected three antibiotics with different modes of action: the cell wall synthesis inhibitor ampicillin, the protein synthesis inhibitor chloramphenicol, and the DNA replication inhibitor norfloxacin. Four independent lineages of wild-type *E. coli* BW25133 were then allowed to evolve increasing resistance to each antibiotic in minimal medium with either glucose or acetate as the sole carbon source. These two carbon sources impose radically different metabolic states: rapid growth with respiro-fermentative metabolism and slower growth with fully respiratory energy generation (Fig 1A).

Selection of resistant mutants was achieved by serial passage in a 96-well plate cultivation format (Fig 1B). To maintain a constant selective pressure at every passage, each culture was inoculated into seven different drug concentrations. At the end of a 48-h cultivation cycle, cells growing at the highest tolerable concentration were used for inoculation at the next passaging round, until 150–160 generations were reached for all lineages. As controls, two independent culture lines were evolved on glucose and acetate without antibiotics. Despite similar mutation rates on glucose and acetate (Dataset EV1) and previous experimental evidence that slowly growing cells are intrinsically more tolerant to antibiotic stress (Gilbert *et al*, 1990; Claudi *et al*, 2014), resistance to all three tested antibiotics evolved much faster on glucose than on acetate (Figs 1C and EV1), demonstrating that environmental conditions can constrain the rate of resistance acquisition.

To shed light on the underlying mechanisms by which metabolism constrained the path of resistance evolution, we profiled the metabolome of evolved populations by nontargeted mass spectrometry (Fuhrer *et al*, 2011). Seven to eight populations from different

points along the evolutionary trajectory were selected from each of the 24 antibiotic-evolved lineages. The resulting 190 evolved populations were regrown on the carbon source used for selection but without antibiotic addition, while the endpoint of antibiotic-free evolved lineages and the wild-type ancestor *E. coli* strain were grown in both glucose and acetate minimal media (see Materials

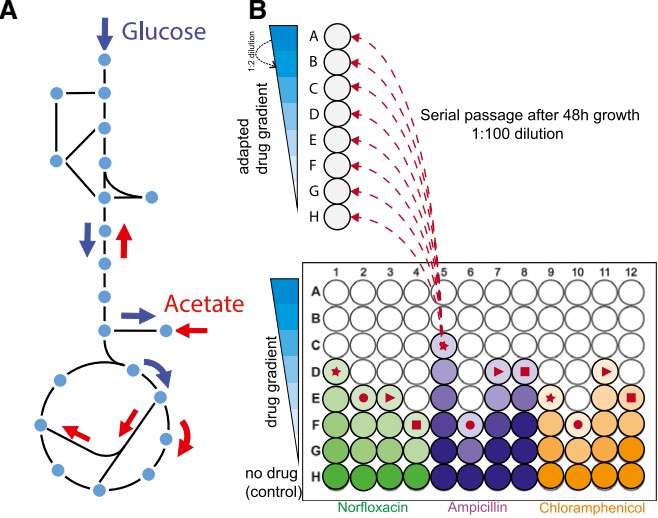

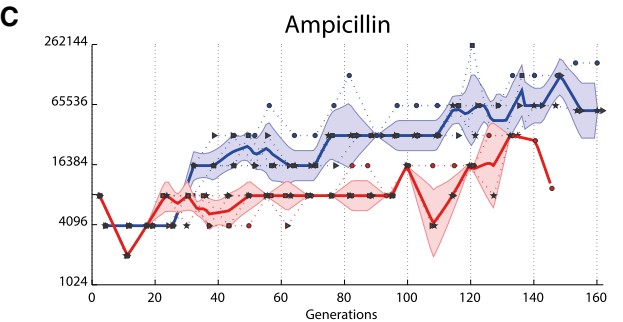

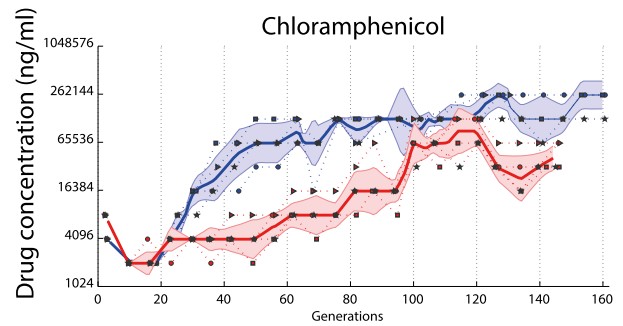

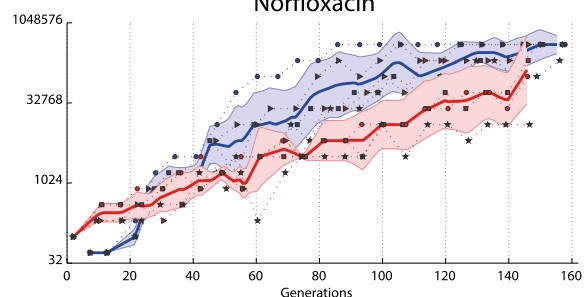

**Figure 1. Evolutionary trajectories of *Escherichia coli* evolving resistance to three different antibiotics on two different media.**

A   Metabolism on glucose and acetate. Glucose is catabolized by glycolysis and can be fermented and/or oxidized via secretion of acetate or tricarboxylic acid cycle (TCA) (blue arrows), respectively. Acetate forces a complete different distribution of internal fluxes and bacterial growth is strictly respiratory (red arrows).

B   Schematic representation of the evolutionary experiment. Each well in a column corresponds to a different dilution of the same antibiotic. Every 48 h, out of the cultures that grew to an $OD_{600} \geq 0.5$, the one that survived the highest antibiotic concentration is selected. Selected population for the next passaging step are indicated by the symbols: ●, ►, ■, ★ indicating the four lineages evolved under the same selective pressure. Selected evolved populations are diluted into eight different antibiotic concentrations, such that at every passaging step 12 populations on glucose and 12 populations on acetate are propagated. At each inoculation step, the highest drug concentration tested was adjusted to be at least double of the concentration where bacterial growth was detected in the previous passaging step.

C   Evolution of resistance. Each dot (●, ►, ■, ★) corresponds to one evolved population selected during the serial passage experiment. *Y*-axis indicates the antibiotic concentration at which evolved populations were selected during serial passages (blue, glucose; red, acetate). Solid line: median of the four lineages, dotted line: single lineages, shaded region is median ± standard deviation across the four lineages.

and Methods for full details) populations. Intracellular and extracellular samples were taken during steady-state exponential growth, and relative abundances of 413 intracellular and 392 extracellular ions, that based on measured accurate mass could be putatively matched to 586 and 553 deprotonated metabolites, were measured by time-of-flight mass spectrometry (TOF-MS; Dataset EV2).

Both the intra- and extracellular metabolome underwent drastic changes after only a few generations (Fig 2A and B), and the changes were highly reproducible across lineages evolved under identical selection pressure (Figs 2A and B, EV2 and EV3). It is worth noting that antibiotic-resistant populations exhibited generally altered metabolomes when compared either with the ancestral strain or with populations evolved without antibiotic selection. This observation suggested that a large portion of metabolic adaptive changes in antibiotic-resistant populations is driven by the respective antibiotic, and not by adaptation to the carbon source (Figs 2B, EV2 and EV3).

Metabolite changes that are the result of adaptive mutations are expected to (i) exhibit clear monotonic transitions from wild-type basal concentrations to new different steady-state levels and (ii) to occur reproducibly across the four lineages evolved under the same selective pressure. We discriminated such metabolic adaptations from transient or stochastic fluctuations by fitting a sigmoidal curve to the relative metabolite abundances in the evolved populations from each lineage. The quality of fit (adjusted $R^2$) was used to systematically identify metabolites transitioning to a new steady state (Dataset EV3 and Fig 2C). Overall, a tendency for larger metabolic rearrangements was observed in glucose-evolved cultures (Fig 2D), where the metabolic changes were also more homogeneous across evolutionary lineages. The faster rate of resistance evolution and the extent of metabolic rearrangement on glucose presumably reflect the higher degree of freedom for respiro-fermentative catabolism, and thus the larger potential for metabolic compensation of antibiotic resistance (Appendix Fig S1). Further interpretation of metabolite changes required development of new unbiased network-based methods for data analysis, since changing metabolites rarely clustered within canonical pathways (Appendix Fig S2).

### A constraint-based modeling approach to interpret evolutionary metabolic adaptation

To functionally interpret the evolved metabolic states in antibiotic-resistant cells, we used concepts derived from flux balance analysis

(Fong & Palsson, 2004; Pál *et al*, 2006) and interrogated metabolomics data with a genome-scale model of *E. coli* metabolism (Orth *et al*, 2011). Classical flux balance analysis is a powerful constraint-based approach to model steady-state internal fluxes (Fong & Palsson, 2004; Orth *et al*, 2011) given a stoichiometric matrix of metabolic reactions and a cellular objective function. Since microbial metabolism is shaped by multiple competing objectives, such as minimization of proteome resource investment (Lewis *et al*, 2010), adaptability to sudden environmental changes (Schuetz *et al*, 2012), maximization of biomass production (Fong & Palsson, 2004) or energetic efficiency (Schuetz *et al*, 2007), natural evolution is expected to select the best tradeoff between these competing objectives, such that metabolism operates in the proximity of a so-called Pareto front (Schuetz *et al*, 2012). However, evolution of resistance to antibiotics introduces new constraints and objectives, making it unclear which potentially new objective functions shape metabolic adaptation (Appendix Fig S3).

While flux changes can in principle result from changes in enzyme abundance or mutations affecting kinetic parameters of the reaction (e.g. $K_{cat}$, $K_m$ values), empirically we observed that a change in flux is often accompanied by changes of metabolites abundance (Boer *et al*, 2010) and that adjustments of enzyme abundance alone are often insufficient to explain flux changes (Fendt *et al*, 2010; Chubukov *et al*, 2013; Reznik *et al*, 2013; Gerosa *et al*, 2015). Hence, we used the reverse approach by assuming that altered metabolite concentrations in evolved strains reflect an attempt to redirect intracellular fluxes toward specific but unknown metabolic objectives to drive and compensate for resistance. To test for this possibility, we systematically minimized or maximized fluxes through each individual reaction of the *E. coli* genome-scale metabolic model (Orth *et al*, 2011). For each reaction, we calculated the shadow prices (Reznik *et al*, 2013; Appendix Figs S4 and S5), which estimate the sensitivity of the objective function (i.e. reaction flux) to changes in the availability of all individual metabolites (see Materials and Methods for full details). Metabolites with negative shadow prices can be interpreted as limiting quantities for the reaction. Next, we used a permutation test to select reactions where metabolites with negative shadow prices were significantly (*P*-value ≤ 0.001) overrepresented among metabolites experimentally found to be altered during evolution of antibiotic resistance. By using shadow prices to interpret measured metabolite level changes, we implicitly assume that the newly evolved flux states will be reflected in altered steady-state concentrations of metabolites that are limiting

---

**Figure 2. Metabolic rearrangements during acquisition of antibiotic resistance.**

A  Pairwise similarity between metabolite profiles of populations that evolved resistance to ampicillin on glucose. Spearman correlation (Fieller *et al*, 1957) is used to assess the pairwise similarity between *Z*-score normalized metabolite changes. Selected populations are indicated by (i) three letters indicating the selective pressure, in this case ampicillin (AMP), (ii) followed by evolutionary lineages, referred to as lineage 1–12, where 5–8 evolved resistance to ampicillin, and (iii) number of generations (Dataset EV2).

B  Pairwise similarity between metabolome profiles of evolved populations. Spearman correlation (Fieller *et al*, 1957) is used to assess the pairwise similarity between *Z*-score normalized metabolite changes in the 193 selected mutants. Yellow bars on the side indicate the wild-type ancestor and the two populations evolved in glucose and acetate antibiotic-free media. For a given drug, all selected populations of one lineage from the evolutionary experiment are in consecutive order and all four lineages are displayed one after another.

C  Intracellular pantothenate levels in ampicillin-resistant *Escherichia coli* populations. Values are normalized to the wild-type ancestor. For the populations belonging to each of the four independently evolved lineages, a sigmoidal curve is fitted and the resulting adjusted sum of squared errors ($R^2$) is reported. Data are the mean ± standard deviation across biological replicates.

D  Metabolic rearrangements. For each independently evolved lineage, the number of metabolites with an adjusted $R^2$ from the fitting analysis greater or equal to an arbitrary stringent threshold of 0.6 is reported for intracellular and extracellular metabolites.

E  Distribution of predicted EMC across metabolic pathways. For each pathway, the relative percentage of EMCs is reported.

---

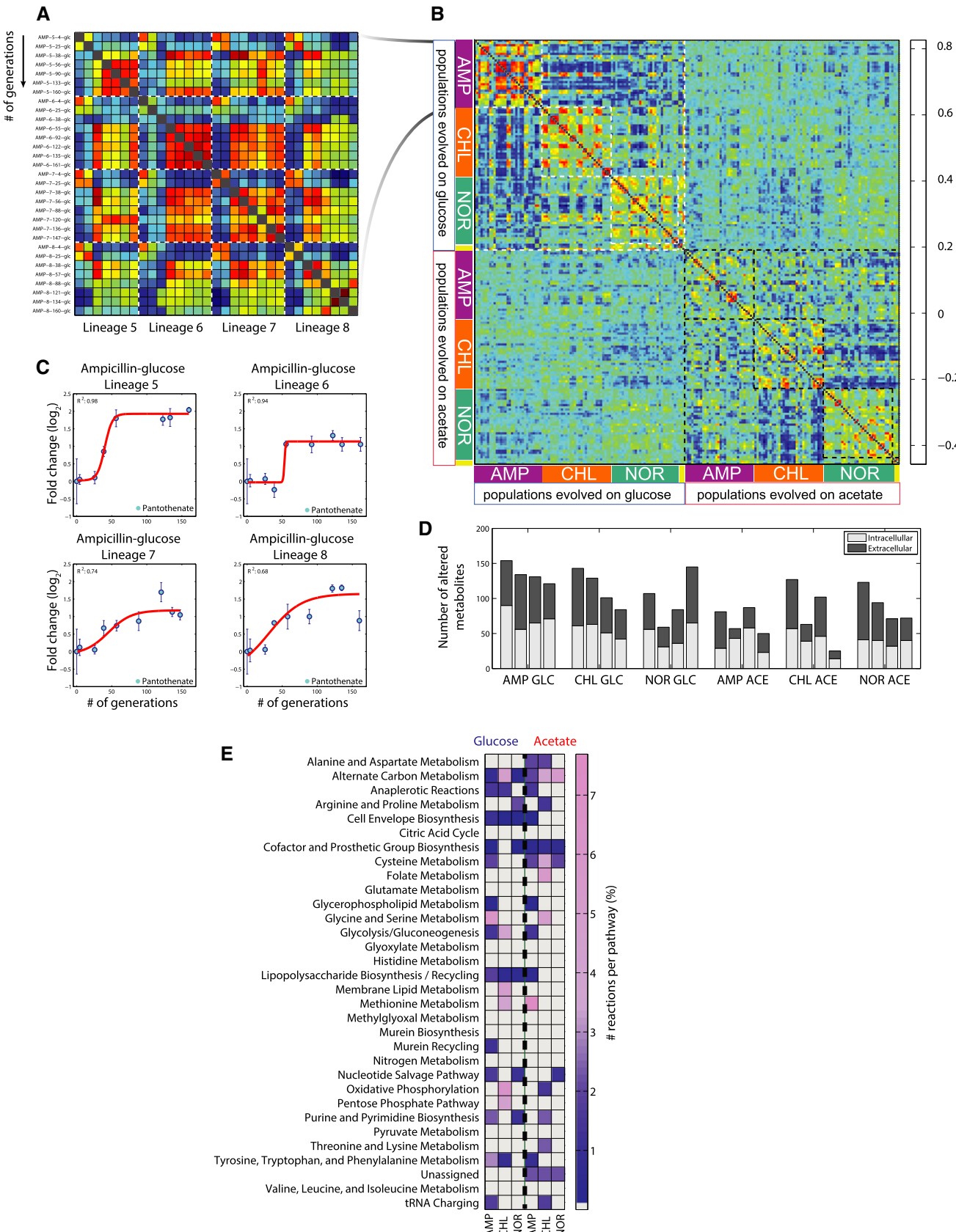

Figure 2.

for the evolved metabolic functions. By systematically searching for reactions with an overrepresentation of altered limiting metabolites, we thus try to identify metabolic functions that if modulated can play an active role in the evolution of resistance or its compensation. We refer to these reactions as evolved metabolic characteristics (EMCs; see Dataset EV4 for full list).

The above metabolic rearrangements were quantified in the absence of antibiotics to ensure that they reflect the evolved compensatory adaptations of resistant *E. coli*, rather than their immediate stress response in actually dealing with the different antibiotics themselves. Next, we asked whether the evolved metabolic traits could be functionally related to the direct effects of inhibition of antibiotic targets. To this end, we monitored the short-term metabolic response of wild-type (antibiotic-sensitive) *E. coli* grown in glucose minimal medium, 1 h after treatment with the respective antibiotics (Appendix Fig S6). The basic premise is that metabolic changes directly induced upon antibiotic treatment reflect the rapid metabolic adaptation to inhibition of the drug target (e.g. gyrase upon norfloxacin treatment). The direct antibiotic response of intermediates in some metabolic pathways of wild-type *E. coli* was similar to the persistent response in evolved populations in the absence of the antibiotic. These constitutive metabolic rearrangements were often independent of the nutrient environment used during selection, such as changes in steady-state levels of intermediates in nucleotide metabolism across norfloxacin-evolved populations (Fig 2E). These relatively few common changes might relate to mutations directly affecting the function of drug targets, such as mutations within the gyrase complex.

The majority of EMCs, however, were in metabolic pathways not directly affected by an antibiotic treatment. Hence, most metabolic phenotypes in evolved *E. coli* reflect resistance or compensatory mechanisms involving metabolic processes not directly affected by the short-term action of the antibiotics. These EMCs unveiled unexpected and radical differences in metabolic adaptation to a given antibiotic as a function of the carbon source used during selection. This environmental influence was most evident in chloramphenicol- and ampicillin-evolved populations, which also exhibited the largest metabolic changes throughout evolution (Fig 2D). Thus, the detected EMCs represent stable evolved traits that can be expected to reflect compensatory and resistance mechanisms that could have not been predicted from the specific antibiotic response.

### Rearrangements of central carbon metabolism in evolved *E. coli* under chloramphenicol and glucose selective pressures

Predicted EMCs in chloramphenicol-glucose-evolved populations involved sugar transport, oxygen uptake, and CoA formation, suggesting major changes in glucose catabolism (Fig 3A). Indeed, the evolved populations exhibited higher rates of glucose consumption, acetate secretion, and a reduced relative oxygen uptake, revealing a switch from respiratory to fermentative metabolism (Fig 3B and Dataset EV5). Surprisingly, anaerobically growing wild-type *E. coli* showed an increased susceptibility to chloramphenicol (Fig EV4), raising the question of how increased aerobic fermentation confers advantages in chloramphenicol-resistant mutants. To this end, we performed genome sequence analysis of the 26 cell populations at the endpoint of evolution, and used multivariate statistical analysis to identify putative mutated genes whose

presence correlated with measured respiration rates in glucose-evolved mutants (Dataset EV6). The mutations in four genes, *acrB*, *acrR*, *fecA*, and *yjhF*, were significantly associated with reduced oxygen uptake ($P$-value $\leq 0.01$; Dataset EV7), and in particular, mutations in the promoter region of the multidrug efflux pump encoding *acrB* were the most significant ($P$-value = 0.001). Consistently, we showed that: (i) chloramphenicol treatment selectively induces increased transcription of *acrB* (Appendix Fig S7), (ii) AcrAB efflux pump is essential to cope with chloramphenicol, as the deletion of the respective encoding genes render *E. coli* much more sensitive to chloramphenicol (Nichols *et al*, 2011; Appendix Figs S8 and S9), (iii) deletion of the efflux pump repressor genes *marR* or *acrR* caused a strong increase in glucose fermentation via acetate secretion (Fig EV5), and (iv) chloramphenicol-resistant populations evolved in glucose in the absence of the antibiotic constitutively exhibited almost four times higher AcrB protein levels than any other evolved population and wild type (Fig 3D).

How could efflux pump overexpression cause the metabolic switch? Generally, overexpression of membrane proteins in *E. coli* leads to a down-regulation of the tricarboxylic acid cycle (Wagner *et al*, 2007) and causes a shift to fermentative metabolism, which may result from a reduced metabolic proteome allocation (Hui *et al*, 2015; Appendix Fig S10) or competition for membrane space with oxidative phosphorylation proteins (Zhuang *et al*, 2011). Since acetate metabolism depends on respiration, a similar compensatory mechanism during growth on acetate would have more drastic consequences on cellular fitness. Interestingly, a functional link between chloramphenicol resistance and membrane proteome remodeling comes from phenotypic profiling of the *E. coli* gene deletion library where many chloramphenicol-resistant mutants were more sensitive to cell wall-damaging agents (e.g. ampicillin or oxacillin) or oxidative phosphorylation inhibitors (e.g. carbonyl cyanide *m*-chlorophenyl hydrazone (CCCP) or theophylline), and vice versa (Nichols *et al*, 2011; Appendix Fig S11). Albeit circumstantial, this evidence suggests the balance between oxidative phosphorylation activity and the membrane composition as an important constraint during evolution of antibiotic resistance, which deserve more attention in future studies.

### The role of cell wall recycling in mediating ampicillin resistance

In ampicillin-resistant populations, we identified major EMCs on glucose in nucleotide metabolism, serine biosynthesis, and cell wall recycling (Fig 4A). We focused on the anhydromuropeptide transport in cell wall recycling (highlighted in Fig 4A and B) because of its proximity to the actual ampicillin target: peptidoglycan biosynthesis. Our EMC predictions (Fig 4A) based on metabolite changes in resistant populations suggested recycling of anhydromuropeptides to play an important role in mediating resistance to ampicillin. In support of this hypothesis, we demonstrated ampicillin-glucose-evolved populations to be two to eight times more sensitive to fosfomycin (FOSF; Fig 4C), an inhibitor of peptidoglycan biosynthesis and the last enzymatic step of the anhydromuropeptide recycling pathway (Fig 4B). In contrast, ampicillin-resistant populations evolved in acetate exhibited a similar or higher tolerance to fosfomycin (Appendix Fig S12). The reason why recycling of anhydromuropeptides evolved only on glucose could be the requirement of the glucose PTS phosphotransfer protein EIIA$^{Glc}$ for the activation of

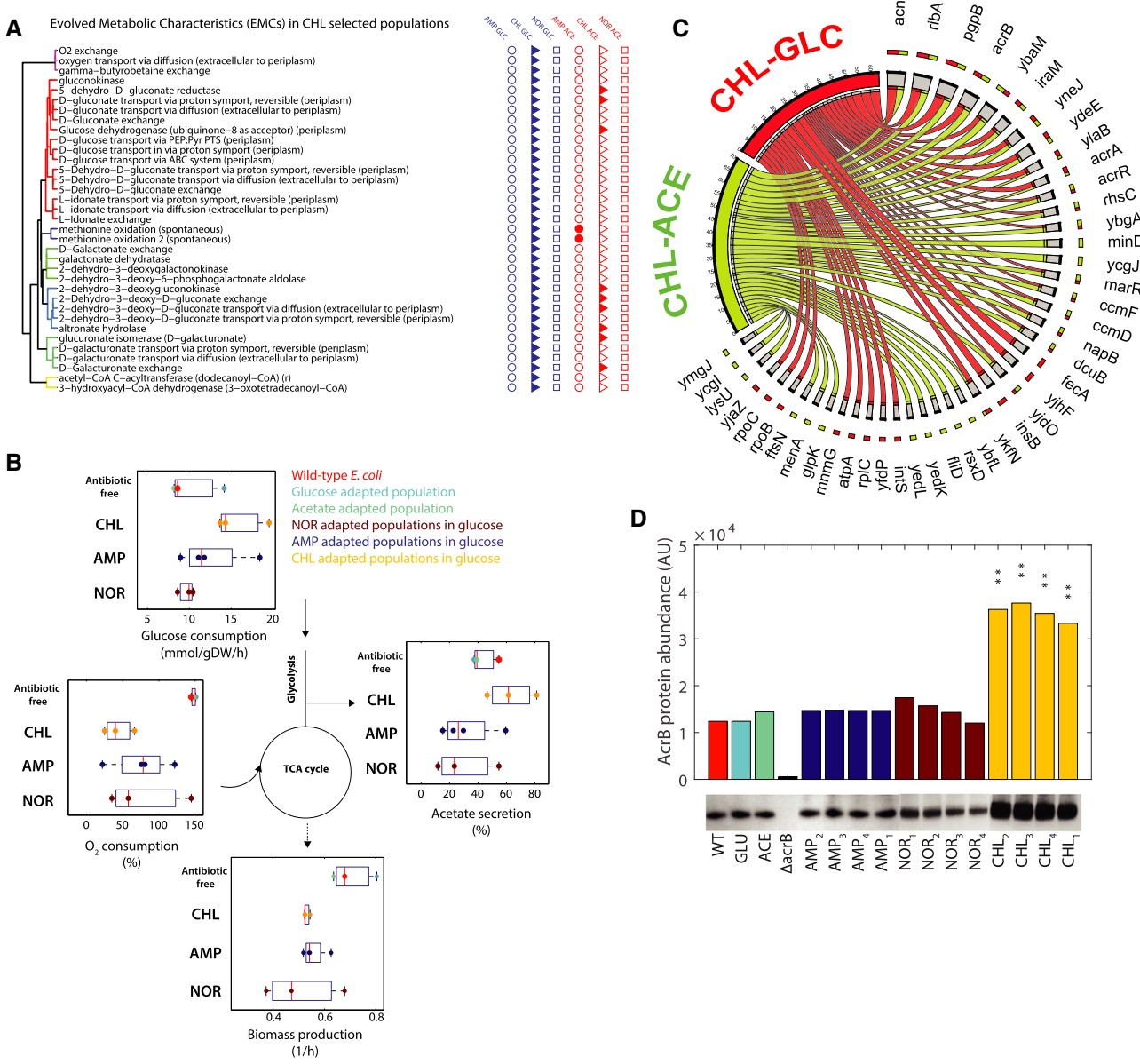

**Figure 3.  Functional metabolic rearrangements in chloramphenicol-resistant populations.**

A   List of EMCs predicted in chloramphenicol-glucose-evolved mutants. Reactions are grouped on the basis of their topological distance, by means of the minimum number of connecting reactions on the metabolic network. For EMCs predicted in chloramphenicol-glucose, filled marks on the right-hand side highlight whether the same EMC was found also in the other evolved populations.

B   Experimentally measured fluxes exclusively in evolved populations grown in glucose minimal medium. Absolute glucose consumption is reported in mmol/gDW/h, growth rate in h$^{-1}$. Acetate secretion and oxygen consumption rates are reported as a percentage relative to glucose uptake. Data have been grouped according to the selective pressure and for each group. The tops and bottoms of each box are the 25$^{th}$ and 75$^{th}$ percentiles of the samples, respectively, while the red line in the middle of each box is the sample median (Dataset EV5 contains mean $\pm$ SD of three biological replicates).

C   Genetic changes identified by whole-genome sequencing. Genetic changes identified in at least two out of the four lineages evolved under the same selective pressure are retained. The bipartite graph links selective pressures (i.e. chloramphenicol-glucose and chloramphenicol-acetate) to mutated genes. Arrow size represents the number of lineages with at least one sequence change in the corresponding gene or its upstream regulatory sequence.

D   Western blot analysis monitoring the AcrB protein abundance across antibiotic-resistant populations evolved in glucose, wild type and populations evolved in glucose and acetate without antibiotics. Asterisks indicate statistically significant difference from wild type *E. coli* (**P < 0.01 from *t*-test analysis). Data are the mean $\pm$ SD of two replicates. One of the Western blots is shown.

MurP, a key protein in N-acetylmuramic acid transport. Expression of EIIA$^{Glc}$ is repressed in acetate minimal medium (Oh *et al*, 2002), which would explain why a similar EMC did not emerge in cultures

evolved under the combined selective pressure of ampicillin and acetate. It is worth noting that differently from acetate, glucose-evolved populations had very few consistent mutations across the

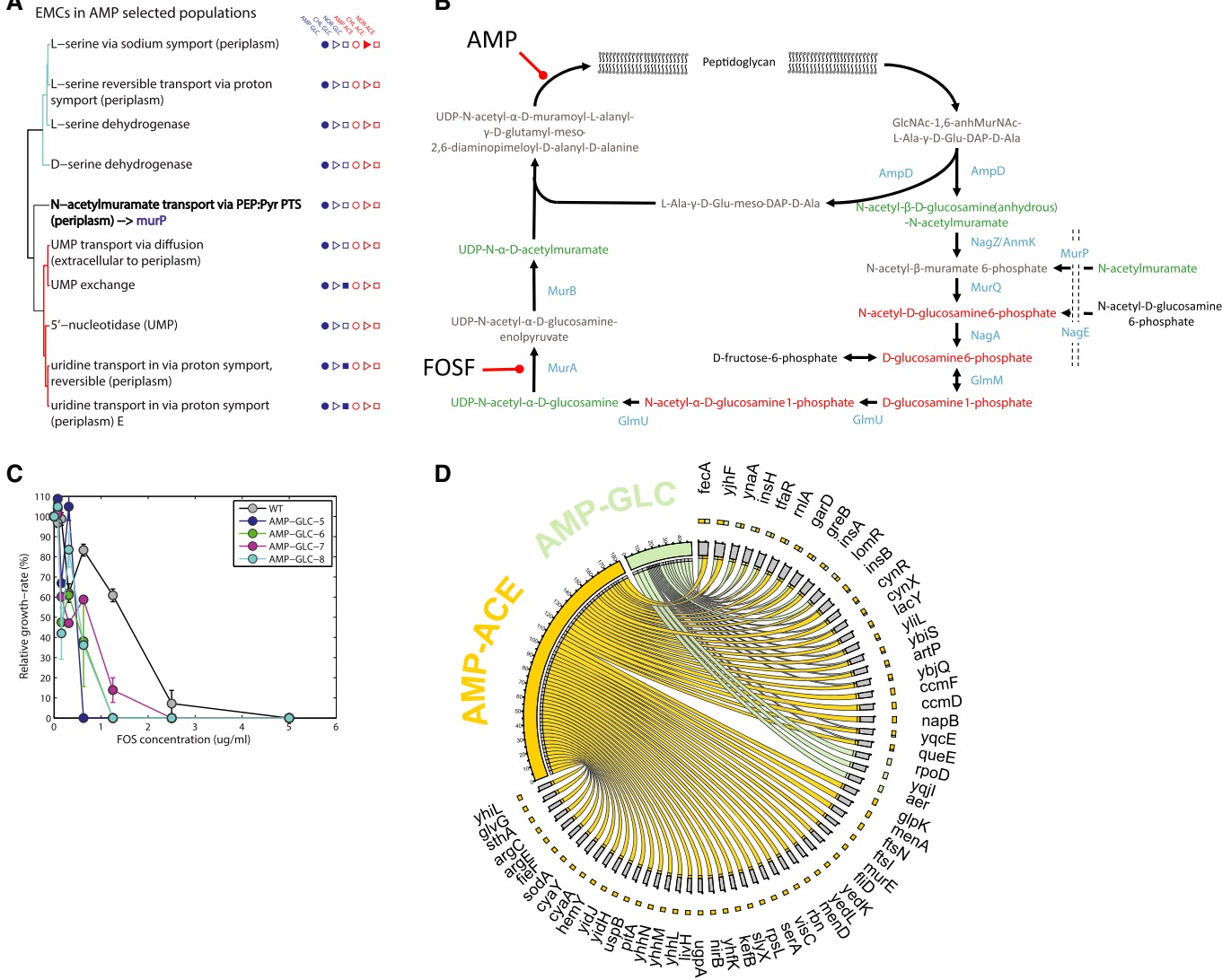

**Figure 4.  Functional metabolic rearrangements in ampicillin-resistant populations.**

A    List of EMCs predicted in ampicillin-glucose-evolved populations. Reactions are grouped on the basis of their topological distance, by means of the minimum number of connecting reactions on the metabolic network. EMCs detected in other evolved populations are highlighted by filled marks on the right-hand side.

B    Schematic representation of cell wall recycling pathway in *Escherichia coli*, adapted from Gisin *et al* (2013). Detected metabolites are highlighted in red or green according to a significant accumulation or depletion in AMP-evolved populations.

C    Sensitivity analysis of ampicillin-glucose to fosfomycin (FOSF). The relative growth rate inhibition of different FOSF concentrations relative to antibiotic-free growth is reported for wild-type (WT) and the populations evolved in the presence of ampicillin and glucose. Data are the mean ± SD of three biological replicates.

D    Genetic changes identified by whole-genome sequencing. Genetic changes identified in at least three out of the four lineages evolved under the same selective pressure are retained. The bipartite graph links selective pressures (i.e. ampicillin-glucose and ampicillin-acetate) to mutated genes. Arrow size represents the number of lineages with at least a mutation (e.g. SNP) in the corresponding gene.

four lineages, mainly affecting regulatory proteins involved in stress response upon environmental changes (e.g. RpoD, Aer, YqjL; Fig 4C), from which no obvious link to anhydromuropeptides recycling could have been made. Hence, directly monitoring metabolic rearrangements in resistant *E. coli* populations was crucial to find the new adaptive mechanisms.

Differential propagation of resistance and compensatory mutations under different nutrition conditions (Figs 3C and 4D) reflect the potential role of metabolism in rendering certain mutations less

accessible by natural selection. Ampicillin-resistant populations evolved in acetate, despite almost four times more mutations than ampicillin-resistant populations evolved in glucose (Fig 4D), exhibited relatively modest metabolic rearrangements (Fig EV3). Altogether, our data suggest that in glucose few mutations offer strong selective advantage and cause sensible metabolic rewiring. On the contrary, in acetate the difficulties in rewiring metabolism make similar jumps in the fitness landscape insurmountable, constraining cells to explore less advantageous solutions. Overall,

we observed large genetic differences in particular between ampicillin- and chloramphenicol-evolved *E. coli* in glucose versus acetate medium (Fig 4D). For example, *murE*, a gene responsible for overexpression of ampicillin targets (Gardete *et al*, 2004) and *ftsI*, encoding for a penicillin-binding protein, were mutated only in cells evolved on acetate, while ampicillin-resistant strains evolved in glucose showed frequent mutations in transcription and sigma factors such *rpoD*, *yqjI*, and *aer* (Fig 4D). Similarly, mutations of the ATP synthase component *atpA*, the 50S ribosomal subunit *rplC*, and the two RNA polymerase subunits, *rpoB* and *rpoC*, were identified only in chloramphenicol-resistant strains evolved on glucose.

## Discussion

A large body of evidence exists on mutations that confer antibiotic resistance, but we know only relatively little about how such mutations affect cellular metabolism, either directly or indirectly. Even less well-understood is how metabolism influences evolutionary strategies to acquire antibiotic resistance. In contrast to genetic screens that identify resistance mutations, we investigated here how metabolism accommodates such resistance mutations. While metabolic adaptation is presumably not the only compensatory mechanism to antibiotic resistance, we demonstrated that environmental conditions play a crucial role in determining the tradeoff between cost and benefit of resistance mutations, and consequently in how rapidly a resistant mutant will establish itself within the population. Specifically, resistance to all three antibiotics evolved much more rapidly on glucose than on acetate, suggesting a greater metabolic plasticity during respiro-fermentative metabolism compared to the obligatory respiratory metabolism on acetate. Developing antibiotic resistance, in turn, also drives metabolic adaptations and the underlying compensatory mutations, which in the two conditions were different for ampicillin and chloramphenicol but similar for norfloxacin (Appendix Fig S13). Despite genome sequence data from evolved populations, which might contain clonal variations, and despite a large genetic space of neutral and beneficial mutations that may confer similar resistance phenotypes, all four parallel population lineages evolved under a given selective pressure converged to highly similar metabolic steady states (Figs 2B, EV2 and EV3). These results suggest that one should consider the natural conditions in tissues or body fluids to better understand the role of metabolic constraints in the evolution of antibiotic resistance in a clinical setting.

During the evolution of resistance, metabolism underwent large condition-dependent rearrangements of metabolite concentrations (Fig 2A and B). These metabolic rearrangements may confer a direct advantage to the mutants in a given selective environment, but may also be their Achilles heels if targeted by a second antibiotic, such as fosfomycin in ampicillin-resistant populations. Hence, understanding metabolic adaptation to evolution of resistance and its compensation can suggest nonobvious targets for multidrug therapies to slow down evolution of resistance, or reveal weaknesses that confer hypersensitivity to alternative treatments in evolved resistant bacteria (Lázár *et al*, 2013; Gonzales *et al*, 2015).

The systematic experimental and computational framework developed in this work generates testable predictions on the functional role of metabolic changes upon evolution of antibiotic resistance. Beyond inferring adaptive mechanisms in the evolution of resistance to antibiotics, it can easily be extended to other drug responses or biological systems such as naturally evolved resistant pathogens. These adaptive mechanisms could not have been derived solely from genome sequence analysis of evolved populations, and classical targeted LC-MS/MS approaches would have been prohibitive due to the large sample size and their relatively low coverage. Moreover, our constraint-based modeling approach differs significantly from other metabolome analysis frameworks such as classical pathway enrichment analysis (Subramanian *et al*, 2005) because we explore the entire network topology in a context-dependent manner, such that different environments/nutrients can lead to different predictions of limiting metabolites and hence, different interpretations of metabolome changes. Understanding the relationship between external nutrients and resistance acquisition and compensation has the potential to suggest less conventional strategies to slow down or prevent selection and emergence of resistance mechanisms.

## Materials and Methods

### Bacterial strains and culture conditions

*Escherichia coli* BW25113 was used as the wild-type strain throughout this study. Growth medium was standard M9 minimal medium with 5 g/l of acetate or glucose as carbon sources, in addition to (per liter) 7.52 g $Na_2HPO_4 \cdot 2H_2O$, 3 g $KH_2PO_4$, 0.5 g NaCl, 2.5 g $(NH_4)2SO_4$, 14.7 mg $CaCl_2 \cdot 2H_2O$, 246.5 mg $MgSO_4 \cdot 7H_2O$, 16.2 mg $FeCl_3 \cdot 6H_2O$, 180 µg $ZnSO_4 \cdot 7H_2O$, 120 µg $CuCl_2 \cdot 2H_2O$, 120 µg $MnSO_4 \cdot H_2O$, 180 µg $CoCl_2 \cdot 6H_2O$, 1 mg thiamine·HCl. Chloramphenicol, ampicillin, and norfloxacin were purchased from Sigma.

### Antibiotic evolutionary experiment

For each of the six drug/media combinations, four independent lineages were propagated in parallel. Serial passaging was performed in 96 deep-well plate cultivation (2 ml well volume, 900 µl culture volume; Fig 1B). Seven wells in a plate column were prepared with gradually increasing concentrations of the same antibiotic, and the last row of the plate served as a growth control and contained no drug. Every 48 h, $OD_{600}$ was measured with a plate reader. The bacterial population that was able to grow (i.e. OD ≥ 0.5) at the highest of seven tested drug concentrations was used for the next passaging step. 9 µl of the selected bacterial culture (e.g. surviving to the highest drug concentration) was used for reinoculation. The number of generations during each passaging step was calculated by (i) measuring the final OD after a 48-h growth cycle ($OD_{fin}$), (ii) 9 µl of selected evolved populations was reinoculated in 900 µl of fresh medium yielding a 1/100 dilution for the new starting OD. At the end of the 48-h growth cycle, OD was measured (OD*) and number of generation is calculated by the following formula: $\log_2(OD*/(OD_{fin}/100))$. At each propagation step, an aliquot of the culture was frozen and stored at −80°C to obtain a library of mutants at different stages of the evolutionary experiment. At each reinoculation step, the highest drug concentration tested was adjusted to be at least double of the concentration where bacterial growth was detected in the previous passaging step.

In parallel, a similar experimental setup was used to evolve two independent cultures in antibiotic-free media. Evolutionary lineages on the two growth media, glucose (GLC) and acetate (ACE) M9, are referred to as lineage 1–12, where 1–4 evolved resistance to norfloxacin, 5–8 to ampicillin, and 9–12 to chloramphenicol. Similarly to other laboratory evolution strategies (Toprak *et al*, 2012), this experimental setup allowed us to qualitatively monitor the rate at which evolution of resistance progresses.

**Whole-genome sequencing**

Evolved populations were inoculated from frozen stocks, grown overnight in LB medium, and chromosomal DNA was purified using commercial bacterial DNA isolation kits (QIAGEN DNeasy Blood & Tissue Kit). Isolated DNA was submitted to the functional genomics center Zurich (http://www.fgcz.ch/) for whole-genome sequencing on an Illumina Gene Analyzer II× (75 bp single-end reads, average coverage of 6 million reads per strain). Raw reads were aligned to the reference genome of *E. coli* K-12 BW25133 using bowtie 2 (Langmead & Salzberg, 2012). Duplicated alignments were removed from the alignment files (bam) using samtools v.1.0 (Li *et al*, 2009), and the variant calling pipeline of GATK v3.2 (McKenna *et al*, 2010) was applied to identify mutations. In particular, the HaplotypeCaller was employed and a minimum coverage of 20× was imposed. The vcf variant files were annotated using SnpEff v 4.0 (Cingolani *et al*, 2012). Raw genome sequence data are available at the European Nucleotide Archive (http://www.ebi.ac.uk/ena/data/view/PRJEB 19222).

**Metabolite extraction and profiling**

For each of the four lineages evolved under the three different selective pressures (i.e. ampicillin, chloramphenicol, and norfloxacin) in the two media (glucose and acetate M9), we selected eight different populations at intermediate stages during the evolutionary experiment. Overall, we selected 192 populations under antibiotic selective pressure. Two of these populations did not grow properly during the metabolome sampling experiment (i.e. AMP 5_3 and NOR_4_3 evolved in glucose M9) and were excluded from further analysis. We also profiled the endpoints of evolution for two *E. coli* populations evolved in glucose or acetate minimal medium without any antibiotics, and wild type. The resulting 193 different *E. coli* populations were cultivated in duplicate in 96-well deep-well plates and samples were collected during exponential phase. Evolved populations were grown in antibiotic-free M9 media with the carbon source used throughout the selection process. For sampling, 75 μl of cell culture was transferred to a 96-well storage plate (Thermo Scientific 96-Well Storage Plate) and quenched in a cold ethanol bath at −50°C for 7 s. After quenching, the samples were centrifuged for 2 min at 1,252 *g* and 0°C. The supernatant was discarded and the pellet was extracted with 100 μl of 60% (v/v) ethanol solution at 80°C, for 2 min. Samples were placed at −80°C and stored until further analysis. Eight samples for intracellular metabolite profiling were taken at different time points during exponential growth. Extracellular metabolites were collected by sampling 50 μl of cell culture, diluting 1:4 in 150 μl of water, centrifuging for 5 min at 4,000 rpm and 0°C, and storing at −80°C. Collected samples were directly injected into an Agilent 6550 time-of-flight mass spectrometer (ESI-iFunnel Q-TOF,

Agilent Technologies). Details are described in Fuhrer *et al* (2011). This method is not able to separate compounds with similar $m/z$ and relies on direct ionization without LC separation. To normalize for the so-called "matrix-effect", we extracted *E. coli* cells with the same extraction buffer, and normalize data only within similar nutritional condition (e.g. acetate or glucose minimal media). For 64 metabolites, we could prove that intensities scales linearly with the corresponding metabolite abundance (Appendix Fig S14). Spectral data processing identified 413 intracellular and 392 extracellular ions that based on measured accurate mass could be putatively matched to 586 and 553 metabolites in a genome-scale model of *E. coli* (Orth *et al*, 2011), containing 1,136 unique metabolites.

**Processing of high-throughput metabolome data**

We employed high-throughput time-of-flight mass spectrometry measurements as previously described in Fuhrer *et al* (2011). Intracellular and extracellular metabolome extracts were collected during cell growth from early until late exponential phase. Samples were collected in biological duplicates and arranged in 96-well plates before two direct injections.

Raw data normalization was a critical step to obtain accurate semi-quantitative metabolite concentrations. We considered the impact of (i) plate-to-plate variance, (ii) the intensity drift during sequential injection, and (iii) matrix effects. We modeled linear dependencies between measured ion intensities and (i) drift during sequential injection for each plate, (ii) amount of extracted biomass quantified ($OD_{600}$), and (iii) the total sum of measured ion intensities. For each metabolite $m$, the relative change in sample $u$ ($FC_m$) was calculated as follows:

$$FC_m = \log_2\left(I_m/(\alpha_m OD_u + \beta_{j,m} K + \gamma_{j,m} TIC_u)\right)$$

where $I_m$ is the measured intensity of metabolite $m$, $OD_u$ represents the optical density at the time of extraction of sample $u$, $\alpha$ represents the linear dependency between measured intensities and OD, $\beta_j$ represents the linear dependency of measured intensities with the temporal drift during injections in plate $j$, $K$ is the injection sequence (from 1 to 96, number of wells in the same plate), $\gamma$ is the linear dependency with Total Ion Counts (TIC) in plate $j$. The proportionality factors, $\alpha$, $\beta$, and $\gamma$, were determined by multiple least square fitting analysis for each ion individually across all measured samples. For each evolved population(s), fold-change and variability of metabolite m are, respectively, the average and standard deviation of $FC_m$ across collected samples (Dataset EV2) and biological replicates.

**Metabolome data analysis**

For each annotated metabolite, log₂ fold-changes in evolved populations are calculated with respect to the ancestor strain and reported in Dataset EV2. Relative metabolite changes ($FC_m$) in the eight selected populations belonging to the same lineage are sorted according to the number of generations ($g$). A weighted least square fitting analysis is then used to fit a sigmoidal curve function:

$$FC_m(g) = p_{1,m} + \frac{(p_{2,m} - p_{1,m})}{1 + 10^{(p_{3,m} - g)^* p_{4,m}}}$$

where $p_1$ is the minimum of the function values, $p_2$ is the difference between maximum and minimum, $p_3$ is the number of generations at which metabolite concentration reaches half of its maximum level, and $p_4$ is the slope. Quality of the fitting is assessed by estimating the adjusted $R^2$ values using the MATLAB function "fitnlm", and weights used are the inverse of the fold-change standard deviation. A sigmoidal model intrinsically captures slow and rapid changes from one basal state to a new different state. For those metabolites where a sigmoidal curve exhibits poor descriptive performance (e.g. low adjusted $R^2$) either: (i) data are too noisy, (ii) there are no significant changes in the relative abundance of the metabolite during evolution of antibiotic resistance, (iii) changes are transient and reabsorbed to a normal basal level by the end of our evolutionary experiment.

## Estimation of substrate consumption and byproduct secretion rates

Of the 12 investigated evolved populations, two did not grow to a sufficient OD and are therefore excluded from further analysis (lineages 3 and 12 that evolved resistance to norfloxacin and chloramphenicol). Selected populations were grown in glucose minimal medium, and supernatant samples were collected during exponential phase. Residual glucose was quantified using enzymatic assays kits (Megazyme), and acetate was quantified by high-performance liquid chromatography (HPLC). HPLC analysis did not reveal significant secretions of other plausible fermentation products, such as citrate, succinate, fumarate, lactate, malate, oxaloacetate, or pyruvate. Relative oxygen uptake rates were measured using the Oxygen Consumption Rate Assay Kit (MitoXpress®-Xtra HS Method) following the suggested protocol. An absolute estimate of oxygen consumption was made by assuming the uptake rate of oxygen in wild type equals 14.93 mmol/gDW/h as reported in Covert *et al* (2004). Measured rates are reported in Dataset EV5.

## Shadow Price estimation and EMC inference

The *E. coli* MG1655 genome-scale metabolic model (Orth *et al*, 2011) was used to calculate the shadow prices ($w$) associated with each metabolite ($j$) for the systematic maximization/minimization of flux ($v_i$) through each individual reaction ($i$) in the model. In matrix notation, if the primal problem is formulated in its standard form:

Max $b^T v$
subject to $Sv \leq l$, $v \geq 0$,

the corresponding symmetric dual problem is as follows:

Min $l^T w$
subject to $STw \geq b$, $w \geq 0$,

where $w$ is the vector of dual variables.
In our specific case, the primal FBA problems are for each flux $v_i$:

min/max $c^T v_i$
*s.t.* $Sv = 0$
$v_{min} \leq v \leq v_{max}$

where $c$ is a vector with only one nonzero element corresponding to the flux to be optimized, $S$ is the stoichiometric matrix, and $v_{min}$ and $v_{max}$ are the thermodynamic constraints. Hence, the dual problem can be formulated as follows:

max/min $l_L^T v_{min} + l_U^T v_{max}$
$c^T = w^T S + l_L^T + l_U^T$
$l_L \leq 0$
$l_U \leq 0$

where $w$ are the dual variables associated with the mass balance constraints and $l_L$ and $l_U$ the dual variables to the thermodynamic inequality constraints. The CPLEX LP solver was used to find the corresponding dual solution to the FBA problems. Two sets of calculations were performed, corresponding to the two media conditions (glucose and acetate).

Practically, in a classical FBA analysis, where maximization of growth is assumed, a shadow price corresponds to the change in the biomass flux when one of the mass balance constraints is violated (e.g. metabolite deviating from steady state). Wasting of a metabolite (e.g. secretion) with a negative shadow price would have a negative impact on the objective, and hence decrease biomass production (Appendix Fig S5).

Overall, our procedure explores violation of mass balance constraints to predict the link between metabolite and flux changes. Given the medium composition, the stoichiometry of the system, and measurements of actual metabolic changes in evolved populations, we predicted evolved metabolic characteristics (EMC) as follows. Model-based estimated shadow prices were compared to measured altered metabolites in evolved populations. For each condition, the 5% of metabolites with the highest $R^2$ from the sigmoidal fitting analysis ($\Omega$) were retained and used for the comparison. Results are qualitatively similar if the top 1% of metabolic changes are retained (Dataset EV4). It is worth noting that shadow prices are not by any means predictive of metabolite levels, but identify limiting metabolites for specific metabolic reactions, providing a concept to transform the experimentally determined metabolite concentration changes upon evolution of antibiotic resistance into a network of potential flux rearrangements. We focus here on the negative signed shadow prices mostly because we are interested in the concept of limiting resources and how these resources can constrain/shape evolution of metabolism in antibiotic-resistant *E. coli*. A positive shadow price would biologically mean that the metabolite is not a limiting resource for the objective reaction, but rather a toxic element. Moreover, the directionality of the metabolite changes at steady state (e.g. accumulation/depletion) is not discriminative in such a framework. For example, the same higher demand for a metabolite can induce two radically different scenarios: its overproduction and possibly accumulation, or an increased utilization, resulting in decrease metabolite levels. In both cases, the metabolite can still be limiting. Hence, the sign of the measured metabolic changes was not taken into account to establish the link with shadow prices.

For each objective reaction ($i$), the sum of shadow prices for selected metabolites is divided by the total sum of shadow prices. This results in a unique similarity score associated with maximization or minimization of flux $i$:

$$SH_o^i = \frac{\sum\limits_{j \in \Omega} w_j}{\sum\limits_{j} w_j}$$

where $SH_o^i$ denotes the observed statistics associated with maximization or minimization of reaction $i$, given the set of altered metabolites $\Omega$. To avoid any assumption on the underlying background distribution and independence of categories, we tested the significance of the observed statistics using a permutation test. Associations between shadow prices and metabolites are randomized 1,000 times, yielding for each tested reaction 1,000 permuted statistics ($SH_P$). Score significance is assessed as follows:

$$P\text{–value} = \frac{\sum\limits_{1}^{1,000} (SH_P \geq SH_o)}{1,000}$$

For each reaction, the lowest *P*-value between maximization and minimization is retained. Reactions identified to be significant (i.e. *P*-value ≤ 0.001) in each of the six tested conditions are reported in Dataset EV4.

### Measurement of *acrB* expression

We used a GFP transcriptional reporter in which the promoter region of *acrAB* was fused to GFP in the plasmid pMW82 (Blair *et al*, 2015a). Expression of *acrAB* was measured during mid-logarithmic phase in M9 minimal medium. Changes in promoter activity during growth were monitored using a plate reader recording GFP intensity and optical density. GFP levels were normalized dividing them by the corresponding optical density.

### Measure of acrB protein abundance

Bacterial samples required for Western blotting were grown aerobically overnight in M9 minimal medium at 37°C. The following day cultures were subcultured and grown in M9 minimal medium at 37°C to approximately mid-logarithmic growth phase ($OD_{600nm}$ ~0.6) then harvested by centrifugation, and cell pellets were re-suspended in 50 mM Tris–HCl (pH 8.0). Protein extracts were prepared by sonication on ice with an MSE Soniprep 150 (Sanyo, UK) for four pulses of 30 s with a 30-s pause between each pulse. A Bradford assay was carried out to quantify the protein concentration, and 10 μg of protein was run on 4–12 % NuPAGE® Bis-Tris mini gels with NuPAGE® MES SDS running buffer (Life Technologies, UK). Protein was transferred to nitrocellulose transfer membranes (Whatman, UK), and analyzed by Western blotting using AcrB antibody at a 1:1,000 dilution. Blots were developed using anti-rabbit IgG horseradish peroxidase-linked antibody (Sigma, UK) at a 1:25,000 dilution and analyzed using the ECL detection system (GE Healthcare UK).

### Elementary flux modes

There are several metabolic operational modes that *E. coli* can explore to grow on a glycolytic substrate like glucose, relatively to a gluconeogenic one, like acetate. For example, cells using glucose can grow in a completely anaerobic environment, avoiding any usage of the TCA cycle, or can redirect carbon to the pentose phosphate pathway to bypass upper glycolysis. On the contrary, *E. coli* is forced to oxidize acetate using the glyoxylate shunt, and gluconeogenesis to feed carbon into pentose phosphate pathway. A systematic analysis of the degrees of freedom in *E. coli* metabolism for respiro-fermentative catabolism of glucose, compared to oxidation of acetate, can be systematically estimated by calculation of all the so-called elementary flux modes.

Elementary flux modes (EFM) are defined as the minimal reaction sets that are able to operate at steady state (Schuster & Hilgetag, 1994). We used the FluxModeCalculator algorithm (van Klinken & Willems van Dijk, 2016) to exhaustively estimate all possible EFMs that can potentially support growth of *E. coli* when growing in a glucose versus acetate minimal media, using the stoichiometric model of central carbon metabolism (http://gcrg.ucsd.edu/Downloads/EcoliCore).

We compared the different EFMs solutions (e.g. set of reactions) that *E. coli* can exploit in order to generate energy to sustain growth and synthetize all the precursors essential for biomass generation when using glucose or acetate. While there are only 506 EFMs that can generate biomass from acetate, we found 83,601 EFMs that can use glucose in order to sustain growth (Appendix Fig S4).

### Metabolic changes upon exposure to norfloxacin, chloramphenicol, and ampicillin

An isogenic strain of *E. coli* BW25113 was grown in glucose M9 minimal medium. Culture volumes of 5 ml were incubated at 37°C, and growth was followed via absorbance at 600 nm. When cell culture reached an $OD_{600}$ of 1, cells were perturbed with the immediate addition of 200 ng/ml norfloxacin, 50 μg/ml chloramphenicol, and 50 μg/ml ampicillin. All samples were harvested after 1 h from drug exposure. Culture broth samples were transferred on a filter, supernatant was fast filtered, and metabolome was immediately extracted. To normalize the amount of biomass extracted, a total Volume × $OD_{600}$ equal to 1 was maintained throughout all samples. The same mass spectrometry technique described in the "Processing of high-throughput metabolome data" section was used. Relative fold-changes were calculated for each of the 437 detected metabolites (Dataset EV2) and significance of the changes calculated by means of *t*-test analysis over three biological replicates. *P*-values were corrected for multiple test by means of *q*-value correction (Storey, 2002). Overall, we found 16, 96, and 55 metabolites with an absolute fold-change > 0.5 and *q*-value lower or equal than 0.01. We calculated the number of significantly changed metabolites within each metabolic pathway relative to the total number of significant changes (Appendix Fig S9). This experiment revealed the metabolic changes induced in wild-type *E. coli* after sudden exposure to the antibiotics used as selective pressures during the evolutionary experiments.

While several affected metabolites were in common among multiple antibiotic perturbations, we observed that the largest fraction of significant changes upon norfloxacin exposure is locating in pyrimidine metabolism, possibly reinforcing the adaptive functions of metabolic changes in nucleotide metabolism upon evolution of resistance to norfloxacin. Similarly, we observed metabolic changes in glycerolipid metabolism only when cells were confronted with ampicillin.

**Possible advantages conferred by higher fermentative metabolism in response to chloramphenicol**

We considered here other possible mechanisms conferring higher tolerance to chloramphenicol upon a reduced cellular respiratory activity.

*Membrane permeability*
Mutations of genes related to the respiratory chain were recurrently observed in aminoglycosides-resistant mutants (Lázár *et al*, 2013). However, differently from aminoglycosides, requiring proton motive force (PMF) to be imported, chloramphenicol can diffuse through the membrane.

*Enzyme cost*
TCA cycle enzymes are among the most costly enzymes for cells (Appendix Fig S15); hence, reduced respiration might compensate for limited proteome resource. Nevertheless, enzymes in TCA cycle occupy a small fraction of the total proteome in the cell, ~5% (Li *et al*, 2014). Hence, it is unclear whether this fraction is enough to justify the metabolic phenotype observed upon deletions of inhibitors of efflux pumps (i.e. *marR* and *acrR*; Appendix Fig S14).

*Oxidative stress*
Recent studies suggested TCA cycle imbalance to aggravate antibiotic toxicity through generation of reactive oxygen species (ROS; Foti *et al*, 2012). However, cells growing anaerobically did not show higher tolerance to chloramphenicol, but rather increased sensitivity, suggesting oxidative stress not to be the driving force of metabolic adaptation in chloramphenicol-resistant populations (Appendix Fig S10).

**Estimation of mutation rate**

To estimate the mutations rates in a glucose and acetate minimal media, we performed a typical fluctuation test (Luria & Delbrück, 1943). 20 cultures of *E. coli* were grown for 48 h in a glucose and acetate minimal media. Cells aliquots were plated on LB agar plates with and without 50 μg/ml of chloramphenicol. After 24 h, cells were counted and estimate of mutation rates was calculated using the formula of Luria and Delbrück (1943):

$$\hat{\mu}_0 = \log 2 \frac{-\log(\hat{p}_0)}{Nt}$$

Results are reported in Dataset EV1.

**Data availability**

Metabolome data are provided in Dataset EV2. Raw genome sequence data are available at the European Nucleotide Archive (http://www.ebi.ac.uk/ena/data/view/PRJEB19222).

**Expanded View** for this article is available online.

**Acknowledgements**
This work was supported by an ETH Zurich Postdoctoral Fellowship to M.Z.

**Author contributions**
MZ and US designed the project. MZ, VC and TE performed the experiments, MZ designed the analysis, and MZ and TE analyzed the data. VR and LP performed the AcrB protein quantification. All authors contributed to preparing the manuscript.

**Conflict of interest**
The authors declare that they have no conflict of interest.

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
