## [Review Process File · Molecular Systems Biology]

Metabolic constraints on the evolution of antibiotic resistance

Mattia Zampieri, Tim Enke, Victor Chubukov, Vito Ricci, Laura Piddock and Uwe Sauer

Corresponding author: Uwe Sauer, ETH Zurich

Review timeline:

Submission date:	20 April 2016
Editorial Decision:	31 May 2016
Revision received:	13 July 2016
Editorial Decision:	28 August 2016
Revision received:	18 December 2016
Editorial Decision:	18 January 2017
Revision received:	28 January 2017
Accepted:	31 January 2017

Editor: Maria Polychronidou

Transaction Report:

1st Editorial Decision

31 May 2016

Thank you again for submitting your work to Molecular Systems Biology. We have now heard back from the three referees who agreed to evaluate your study. As you will see below, the reviewers think that the presented findings seem interesting. However, they raise a number of concerns, which should be carefully addressed in a revision of the manuscript.

The reviewers' recommendations are rather clear and constructive so there is no need to repeat the points listed below. Of course, if you would like to discuss any specific points you can contact us directly.

REFEREE REPORTS

Reviewer #1:

In this study, Zampieri et al. aim to address the degree to which bacterial metabolism constrains evolution of antibiotic resistance. To do this, they evolve *E. coli* to become resistant to three different antibiotics under two fundamentally different carbon sources. Then they profile the metabolome of 190 evolved populations with non-targeted mass spectrometry and integrate the results with a metabolic modeling framework in order to facilitate data interpretation. They make use of the concept of shadow prices in linear programming to integrate their metabolome measurements with the outcome of metabolic flux simulations, which represents a novel strategy to interpret metabolomics data. The obtained results, together with genome sequencing of the evolved strains, help the authors to propose condition-dependent compensatory mechanisms leading to antibiotic resistance.

The angle of the study is novel, the driving question is of extreme interest, and the study is well designed. Interestingly (and perhaps not that acknowledged in the manuscript), the authors demonstrate not only that metabolism constrains the evolution of antibiotic resistance, but also that changes in antibiotic resistance in return rewire metabolism (this for me could be a stronger selling point). Yet, and despite my enthusiasm for the

topic and results, there is a number of issues that have to be addressed, and when necessary clarified, to justify part of the data and their interpretation.

Major comments

1) Mistakes in Figures and poor legend description create ambiguity and impinge on many of my comments later:

a) Fig 1B does drug gradient remain the same across all passages? This is what is implied by schematic. What do the arrows, circles, stars and squares represent on plate?

b) Fig 1C: it is impossible to understand from how much drug resistance you start and to how much you evolve to, because it is unclear what is the metric for your y axes. Do you mean by log₂ that for example in Amp that drug concentration goes from 2¹⁰ to 2²⁰ µg/ml? You cannot be starting from 2000 µg/ml (do you mean ng/ml?). Also why not having a log scale, but putting actual values?

c) Fig 2: confusing that panels come from left to right. Not clear what is the relative metabolite values you use to calculate Spearman correlations (see major comment 2). Better to cluster profiles in panels A & B (see minor comment 6). Last sentence in legend of panel A is confusing (there are no numbers 1-4 or 9-12). In corresponding Sup Figs of panel A (S1 & S2) Nor and Amp have been probably swapped. Panel C legend: to where this R² refers to is unclear. Panel D: Is the metric really percentage of reactions per pathway?! Because most have 1-2%, which apart from being very scarce (statistical significance?), it also means that you have 50-100 reactions/pathway... See also minor comment 9 for this panel.

d) Fig 3: schemes (arrows, circles, stars and squares) are not explained in panel A (same for Fig 4). Panel B box plots and key have non-corresponding colors (NOR-adapted strains) and for all antibiotics it is unclear if this is Glc or Ace evolved strains. Legend for panel B talks about only CHL-CLC adapted strains...

2) The main message of this study is on how metabolism constrains evolution of antibiotic resistance. However, throughout the manuscript, it is hard to disentangle the changes that are due to adaptation to the carbon source from those that are due to adaptation to the antibiotic; neither at the genotypic nor at the phenotypic level. For example, it is unclear to what comparisons are made in Figs. 2A-B/S1-2 (are relative metabolic changes calculated relative to ancestral strain?) or how do the glc/ace-alone evolved strains behave? Then in Figs 3 and 4, it's unclear which mutations are causal and which are passenger-this could be done to some approximation by screening the Keio library in these conditions or at least using the Nichols et al. data for this (with the caveat that this data is in LB).

3) Explanation for evolution of CHL resistance during growth in glucose: this part has weaknesses.

First there are inconsistencies. Authors claim: "Since acetate metabolism depends on respiration, a similar compensatory mechanism to evolving chloramphenicol resistance was not feasible." Yet in Fig 3, *arcR* and *marA* have also mutations in acetate evolved strains. Authors should check if mutations on *acrB/marR/acrR* have an impact on CHL resistance during growth on Glc and Ace (btw is not clear in Fig S11 in what growth media the experiment is done- see also minor comment 14).

Second, fermentative vs respiratory metabolism will impact drastically pmf generation, which is the energy source of the AcrAB-TolC pump. Could it be that mutations on negative regulators of the pump, boost up the levels of the pump, and metabolic reorganization in Glc, tunes in the activity of the pump - so it is active only as much as needed. Pumps are usually not very selective on what they pump out (so may be pumping out also necessary metabolites) and come with a high energy cost. There are simple experiments to check for this scenario: a) probe levels (western blots) and activity (Nile red assay) of AcrAB on Glc and Ace-evolved strains containing e.g. *marR* (or *acrR*) and compare it to corresponding Keio mutant (as non-evolved); b) compare contribution of *marR* or *acrR* mutations to CHL resistance in evolved vs non-evolved strains (it is easy to revert the evolved strain to *marR*⁺ or *acrR*⁺ with P1 transductions).

Third, evidence for connection postulated about membrane proteome reorganization and CHL is unclear. The authors state: "Circumstantial support for hypothesis ... comes from phenotypic profiling of the *E. coli* gene deletion library where many chloramphenicol resistant mutants were more sensitive to cell wall damaging agents or oxidative phosphorylation inhibitors". First, it's unclear how authors assess this. Do they look for anti-correlation between these conditions? Fig S12 and S13 do not help to understand how comparison is made (btw Fig S13 does even not exist, and legends for Figs S12 and S13 are the same). It is also non-intuitive why mutants that potentially have an increased fermentative metabolism (important for chloramphenicol resistance) are sensitized to oxidative phosphorylation inhibitors.

4) Norfloxacin results are only discussed in the last sentence of Results. There should be either more extensively

presented (with main or Supp figure) and discussed or the authors should take these results out completely. Added value for including them in the manuscript, as it currently stands, is little to none.

Minor comments

1) Abstract:

a) 2nd line: maybe "interaction" instead of "synergy"

b) "both in terms of rate and mode of resistance acquisition": since most people will think "rate of resistance acquisition" is equal with "rate of mutation", maybe is better to rephrase or specify here.

c) Last sentence: is unconnected with rest of text - it is not clear how this will help to fight antibiotic resistance (btw this is never discussed in any extent in main text - although it would be beneficial)

2) Intro: "typically, such resistance mutations or genes come at a fitness cost that reduces rate of bacterial proliferation": I am not sure if you can make this as a general statement. It's an expectancy, but there is no systematic study addressing it - or if there is please cite it. For example, many of the pump mechanisms of resistance do not change growth rate.

3) Was pH monitored during the evolution experiments? 0.4% Glucose and respiro-fermentative growth will eventually lead to media acidification (which will depend on how media is buffered - but with M9 0.4% Glc, you are on the border of being able to buffer it). According to authors, the evolved strains in Glc seem to secrete even more acetate. Since pH has a dramatic role in the uptake/activity of antibiotics (e.g. decreases uptake/effectiveness of fluoroquinolones but increases that of chloramphenicol) the authors should check the pH (along the growth curve) for at least the final evolved strains.

4) What is the doubling time in Glc vs Ace? With a 48hrs cultivation cycle, it would be interesting to know how many hrs bacteria spent in stationary phase in each case. With stress (stationary phase)-induced mutagenesis playing a role in mutation rate, this could be a factor.

5) Page 3: "slowly growing cells being more resistant to antibiotics": this is a registered knowledge for decades, so citing a paper from 2014 for this seems "inappropriate".

6) Page 4 & Figures 2A/B; Figures S1/S2; if the statement is that lineages evolved in same stress/media behave same, then it would be better to cluster the correlations of metabolic profiles and not put them in consecutive order.

7) The definition of Evolved Metabolic Characteristics (EMCs) is scattered over the methods section, figure legend and main text. Furthermore, it is unclear what precise metabolites were used to calculate the SH for each reaction i (presumably the ones involved in reaction i). It is also unclear how the concept of "negative shadow prices" was included in this formula and whether absolute values were used or not.

8) As a more general comment, referring to the methods section is missing throughout the manuscript. This would be very helpful for reader to know when he/she should read more to understand the underlying assumptions/work for this statement.

9) Page 6: the authors state "Intermediates in some metabolic pathways affected upon direct antibiotic exposure in wild-type *E. coli* featured similar persistent changes in evolved resistant populations, often independent of the nutrient environment used during selection, such as glycerophospholipid and nucleotide metabolism in ampicillin and norfloxacin evolved strains, respectively (Fig. 2D)." The expression "some metabolic pathways" is ambiguous and it remains unclear what % of the EMCs are indeed not affected by antibiotic treatment. Furthermore, the reference for Fig. 2D after this sentence might make one think that the data shown in the figure comes from antibiotic treated bacteria. I'm not sure whether that is the case, due to the "Acetate" and "Glucose" separation, but nothing is mentioned in the figure legend in this respect.

10) Page 6, next sentence: "The majority of EMCs, however, locates in metabolic pathways, which are not directly affected by the respective antibiotics". This implies that previous discussed pathways have to do with the mode of action of the two antibiotics, but to me this not obvious why.

11) Authors observe similar mutation rates at both carbon sources, and evolution experiments are carried for same number of generations. So expectancy is that evolved strains carry same number of mutations. Yet Amp-Ace evolved strains have 4x more mutations than Amp-Glc ones (Fig 4D), and both are quite different from Chl evolved strains (Fig 3C). Authors should comment on this.

12) Page 7: "Indeed the evolved populations exhibited significantly higher rates of glucose consumption, acetate secretion and a reduced relative oxygen uptake ... (Fig. 3B and Table S5)". As no detailed information on the statistical test used to support this statement is provided, the usage of the term "significantly" can be misleading. Additionally, "Estimated fluxes" in the legend of Fig 3 is ambiguous. Estimated by model simulations or calculated from experimental measurements?

13) Please provide more detailed explanation on how the p-values in Table S7 were calculated. Did you correct for multiple testing?

Btw do these mutations co-occur in the same strains? Since some of them work through the same pathways, I cannot really see the added value of having both.

14) Fig S11: not clear how significant the *acrR* effect is. It is certainly not strong. Also clarify in main text that you are talking about wildtype *E. coli*, when describing this Figure.

15) Testing the Ace evolved mutants in Fosfomycin would be a good control to include.

16) The fact that Ace evolved strains in Amp have so many more mutations - many of which as authors point are in pleiotropic transcriptional regulators, but yet they show they least amount of metabolic rearrangement and resistance to Amp, illustrates very well: a) the confinements metabolism puts in phenotype evolution (main message of paper), but also b) the metabolic plasticity may actually buffer for the selective pressure to evolve (genetically mutate). Latter is only briefly discussed in the text and in my opinion warrants more space.

17) Conclusions: "Emerging ..., Achilles heels if they are exposed to other perturbations"; this a typical case of "collateral sensitivity" where a lot of work has been published lately and would be use to incorporate in discussion.

Reviewer #2:

"Metabolic constraints on the evolution of antibiotic resistance," by Zampieri and colleagues investigates a very important question; how does nutritional environment impact the development of resistance to antibiotics. The authors performed adaptive evolution studies, quantified a large number of intra- and extra-cellular metabolites, and used several computational tools in order to sift through the mountain of data and generate hypotheses. Overall, this reviewer believes that the goals of the study are important and there is a lot of potential for this type of work; however, there are numerous concerns and questions that remain, which need to be addressed and these are differentiated as major and minor issues below.

Major.

1. Fig. 1: How were generations calculated? This is an incredibly important element of the evolution experiments that I do not see explained. Some of my below comments will convey its importance.

2. "...resistance to all three tested antibiotics evolved much faster on glucose than on acetate (Fig. 1C), demonstrating that environmental conditions can constrain the rate of resistance acquisition." First, Fig. 1C needs to be explained better in the legend (e.g., what are the blue and red shading from, and how do authors quantify generation). Second, the authors say that resistance evolved faster on glucose than acetate, but I do not see the statistical test the authors applied to make this conclusion? Third, I find it a bit problematic to use a fixed time (48 hr) to transfer the cultures to the next antibiotic-containing plate for media that have vastly different growth-rates. This is compounded by the fact that transfers were made from wells that reached an OD of 0.5 regardless of the media source. What was the maximum density of the glucose-grown cells w/o antibiotic compared to that of the acetate-grown cells w/o antibiotic? Based on the amount of each used as described in the methods (5 g/L), I suspect that the total amount of growth the glucose could support is much higher than that of the acetate. So with using a fixed time and fixed OD cutoff threshold for transfer, I would expect that slower growth-rates and lower potential final densities would bias results (e.g., for acetate populations the wells used for transfer to next round would come from lower antibiotic concentration wells). To be unbiased in this regard, I would think that the transfers should occur at a time dictated by the unstressed control (e.g., when they reach stationary phase) and transfer should occur at a density that corresponds to a fraction of the density obtained for the unstressed control (e.g., MIC50, or concentration where 50% of growth has been inhibited). I think the authors should adjust their conclusions based on these confounds, or acknowledge them and explain why they do not think they matter with regard to their conclusions.

3. Fig 2a: This is data from ~190 different populations from different times in the trajectory of the 24 evolved lineages with controls. This is from populations correct? Not individual colonies from plating of those populations? So it is possible that the data reflects the metabolic characteristics of a population and not of individual resistant mutants?
4. Fig. 2c: y-axis says significantly changed metabolites. What statistical test was used? It read as if these were just the metabolites with a good Rsquared value for the sigmoidal fits. How does that assesses significance?
5. "Indeed the evolved populations exhibited significantly higher rates of glucose consumption, acetate secretion and a reduced relative oxygen uptake, revealing a switch from respiratory to fermentative metabolism (Fig. 3B and Table S5)." So these were populations right, not colonies obtained from plating of populations that grew at specific antibiotic concentrations? Important for next point.
6. "To this end, we performed genome sequence analysis of the 26 strains at the end point of evolution, and used multivariate statistical analysis to identify putative mutations whose presence correlated with measured respiration rates in glucose evolved mutants (Table S6)." Were these 26 colonies or 26 populations? It matters immensely for interpretation of the whole genome sequencing results. In fact, can the authors please be more explicit in their methods as to how they "called" a mutation. It seems that they required there to be a 20X or higher coverage, but what constituted a mutation, what allele frequency did you use? Its important to know this as well as whether colonies or populations were sequenced.
7. "Consistently, we showed that deletion of the efflux pump repressors MarR or AcrR (Nichols et al, 2011) not only rendered E. coli more tolerant to chloramphenicol (Fig. S11-13), but also caused a strong increase in glucose fermentation via acetate secretion (Fig. S14)." This is an interesting result, but it is not the same as repairing the promoter mutation in the evolved strain, and analyzing its metabolism and resistance. To repair the mutations in the evolved strain and perform these experiments would establish causality. The data with the repressor deletions does not convince me of much with the evolved strains. Also, you could take the parent strain and use the same promoter mutation your sequencing data found. That would be much better than the repressor deletions. Experiments with evolved strain assess if the mutation is necessary, experiments using the promoter mutation in the ancestral strain assess if the mutation is sufficient.
8. "In support of this hypothesis we demonstrated ampicillin-glucose evolved mutants to be hypersensitive to fosfomycin (FOS) (Fig. 4C), an inhibitor of the last enzymatic step of the anhydromuropeptide recycling pathway (Fig. 4B)." The authors neglected to mention that the target of FOS (murA) is an essential enzyme in peptidoglycan biosynthesis, not just the recycling pathway. Also, was this done with populations of mutants or individual mutants that had come from colonies on a plate? The authors should devise other means to test the importance of recycling independent of peptidoglycan biosynthesis (perhaps something with ampD or repairing something in the mutants that is responsible for the peptidoglycan recycling).
9. "Differential propagation of resistance and compensatory mutations under different nutrition conditions (Fig. 3C, 4D) reflect the potential role of metabolism in rendering certain mutations less accessible by natural selection." First, why are 3C and 4D generated by having a different threshold of mutations, 2 out of 4 for 3C and 3 out of 4 for 4D? Second, since only 4 populations were propagated for each media and antibiotic combination, how can the authors make this claim? If the mutation is less accessible, the authors should see a statistically significant difference in mutations to specific loci in glucose plus antibiotic when compared to acetate plus antibiotic. Was this statistical comparison made to identify loci (e.g., murE, rpoD, etc.) that are statistically less accessible between conditions?

Minor

1. First sentence of the summary, "While we begin to understand the genetic basis of antibiotic resistance, ...". This is highly inaccurate. Research on antibiotic resistance has been conducted for over 60 years, and the genetic basis for resistance is understood in a vast majority of cases. Antibiotic resistance forms the basis of a lot of micro- and molecular biology, and writing of that field in this way will turn off many people. I would advise adjusting this statement.
2. "A constrain-based modeling approach to...": should be "constraint".
3. Fig. 3b: "Glycolisis" should be "Glycolysis".
4. Fig. 3b: Can the authors please describe what all of the data points and box-plots are here, I don't find the

legend very helpful in this regard.

5. 5. Top of pg 9: rpoD is a sigma factor, not a transcription factor.

Reviewer #3:

The present manuscript explores the important question of metabolic constraints on the evolution of antibiotic resistance. The experimental design is clearly described, but unfortunately, lack of clarity in presentation of the results and shadow price method makes it hard to evaluate the extent of progress on this important problem.

1. Metabolomics relies on direct ionization without LC separation. While this is a very interesting method and can be used to achieve things that cannot be done with regular metabolomics, it is no substitute for LC-MS due to numerous interferences due to in-source degradation, isomers, etc. It seems likely that the number of reported measured metabolites is greater than the number actually measured (indeed, it seems that the authors count single measured ion as if they measured multiple isomeric metabolites) and more importantly that many are mismeasured due to unknown interferences.

2. The way of detecting metabolite changes relies very much on the "smoothness" of the evolutionary trajectory, without taking into account the quantitative extent of the change. Indeed, the quantitative extent of change in metabolite to show nowhere in the main text. It is hard to evaluate the paper without better sense of what the actual data look like.

3. For reasons that are not well explained, the authors look for enrichment of negative shadow prices. As far as I can understand (although I cannot find the definition of the sign anywhere in the paper), in typical use of shadow prices, if one increases a metabolite like PEP, and you get more feasible flux through a downstream reaction like pyruvate kinase, the sign of the shadow price is positive not negative? Why the focus on negative shadow prices? It would be enormously beneficial for the authors to walk the reader through one or more real examples in the main text from data to shadow prizes to conclusions.

4. I do not know why the authors put much of the critical data, like the response of glycolysis to changes in MarR and AcrR in the supplement?

5. The extent of hypersensitivity to FOS is disappointingly modest. FOS concentration is on a linear axis and one strain seems to show only a < 2-fold change? I am not an expert in this area, but it seems like clinical interest would rely on order of magnitude changes?

6. In general, I find figure 2 fails to convey the most important information. Figure 2 should give a good sense of what is metabolically going on, in terms of metabolite concentrations and any measured uptake or excretion fluxes. Instead, we are left with correlations and overprocessed data that is still in the form of hard to comprehend lists. Much preferred to give a better sense of the actual concentration and flux data and then to pull out a few interesting observations to highlight (perhaps ones that relate to the larger shadow price approach?)

7. Similarly, I find Fig 3 and 4 to convey little. The lists of evolved functions are divorced from any sense of the underlying raw data changes and yet are also very hard to read or understand. Figure 3B is much harder to read than normal bar graphs, and it is hard to match the colors to the conditions. Figure 3C/4D seem to me a very poor way to show the mutation information. Normally these type of diagrams are used to show the co-occurrence of different mutations (where every gene can connect every other gene, and the presence or absence of such linkages is informative). Here it seems that it could be much easier to read a table of mutations divided into acetate, glucose, and both? This might also allow some chance to explain the nature of the underlying mutations and the function of the genes?

8. The supplementary methods describing the shadow price calculations (and formulation as a symmetric dual problem) were incomprehensible to me. Better explanation is needed.

9. To end on something positive, Fig 1 is lovely.

Overall, I do think that there is a lot of interest here, if it could be presented much better.

1st Revision - authors' response

13 July 2016

Reviewer #1:

In this study, Zampieri et al. aim to address the degree to which bacterial metabolism constrains evolution of antibiotic resistance. To do this, they evolve *E. coli* to become resistant to three different antibiotics under two fundamentally different carbon sources. Then they profile the metabolome of 190 evolved populations with non-targeted mass spectrometry and integrate the results with a metabolic modeling framework in order to facilitate data interpretation. They make use of the concept of shadow prices in linear programming to integrate their

metabolome measurements with the outcome of metabolic flux simulations, which represents a novel strategy to interpret metabolomics data. The obtained results, together with genome sequencing of the evolved strains, help the authors to propose condition-dependent compensatory mechanisms leading to antibiotic resistance.

The angle of the study is novel, the driving question is of extreme interest, and the study is well designed. Interestingly (and perhaps not that acknowledged in the manuscript), the authors demonstrate not only that metabolism constrains the evolution of antibiotic resistance, but also that changes in antibiotic resistance in return rewire metabolism (this for me could be a stronger selling point). Yet, and despite my enthusiasm for the topic and results, there is a number of issues that have to be addressed, and when necessary clarified, to justify part of the data and their interpretation.

We are very grateful for the comments. The fact that metabolism is also rearranged upon evolution of antibiotic resistance is highlighted more now in abstract and discussion.

Major comments

1) Mistakes in Figures and poor legend description create ambiguity and impinge on many of my comments later:

a) Fig 1B does drug gradient remain the same across all passages? This is what implied by schematic. What do the arrows, circles, stars and squares represent on plate?

At each inoculation step, the highest drug concentration tested was adjusted to be at least double of the concentration where bacterial growth was detected in the previous passaging step. We make this now explicit in the figure legend. We also rephrased the legend to make clear that the circles, squares, stars and triangles represent only the one population (the one growing at the highest drug concentration) for each evolved lineage that is propagated at the end of the 48 hour growth cycle. The same symbols are used in Fig. 1C.

b) Fig 1C: it is impossible to understand from how much drug resistance you start and to how much you evolve to, because it is unclear what is the metric for your y axes. Do you mean by \log_2 that for example in Amp that drug concentration goes from 2^{10} to 2^{20} $\mu\text{g/ml}$? You cannot be starting from $2000\mu\text{g/ml}$ (do you mean ng/ml ?). Also why not having a log scale, but putting actual values?

We thank the reviewer for spotting this mistake, indeed we meant ng/ml . We changed the axis label and corrected the mistake. Log scale yields a better visualization of results.

c) Fig 2: confusing that panels come from left to right. Not clear what is the relative metabolite values you use to calculate spearman correlations (see major comment 2).

We rearranged the figure. Please refer to major comment 2.

Better to cluster profiles in panels A & B (see minor comment 6). Last sentence in legend of panel A is confusing (there are no numbers 1-4 or 9-12). In corresponding Sup Figs of panel A (S1 & S2) Nor and Amp have been probably swapped. Panel C legend: to where this R^2 refers to is unclear.

We rephrased the figure legend and clarified the nomenclature of the evolved populations. Original figures S1 and S2 were redundant with Figure 2B of the main manuscript and were replaced with two new figures (EV1-2) to address point 6. Please see comment 6 for discussion of clustering of metabolome profiles.

Panel D: Is the metric really percentage of reactions per pathway?! Because most have 1-2%, which apart from being very scarce (statistical significance?), it also means that you have 50-100 reactions/pathway... See also minor comment 9 for this panel.

We used a very restrictive cutoff for selection of significant EMCs. Selected EMCs reduce the relatively large number of significantly altered metabolites (in the order of hundreds of intra- and extra-cellular metabolites) (Fig. 2C) to those reactions predicted to be potentially limited by the availability of these metabolites. As shown for example for chloramphenicol or ampicillin (in glucose Fig. 3A and 4A), reactions predicted as EMCs are indeed a relatively low number and Fig. 2 D is not meant to test for significance of the pathway enrichment, but rather to show where these reactions locate in the metabolic network.

d) Fig 3: schemes (arrows, circles, stars and squares) are not explained in panel A (same for Fig 4). Panel B box plots and key have non-corresponding colors (NOR-adapted strains) and for all antibiotics it is unclear if this is Glc or Ace evolved strains. Legend for panel B talks about only CHL-CLC adapted strains...

We rephrased the figure legend accordingly and correct the color for "NOR-adapted strain". We also explicitly state in the figure legend that we focus here only on strains that evolved resistance to chloramphenicol in glucose minimal medium.

2) The main message of this study is on how metabolism constraints evolution of antibiotic resistance. However, throughout the manuscript, it is hard to disentangle the changes that are due to adaptation to the carbon source from those that are due to adaptation to the antibiotic; neither at the genotypic nor at the phenotypic level. For example, it is unclear to what comparisons are made in Figs. 2A-B/S1-2 (are relative metabolic changes calculated relative to ancestral strain?) or how do the glc/ace-alone evolved strains behave?

We performed a Z-score normalization of the raw data prior to calculation of pairwise similarities between ancestral strain and evolved populations. While we cannot completely disentangle the nutrient adaptation from the antibiotics ones, we found that after only a few generations, antibiotic resistant populations exhibited drastically different overall metabolome changes with respect to both: the ancestral strain and the evolved population without the antibiotic selection. We now explicitly addressed this point in new EV Fig. S1-S2. In this new analysis we followed the reviewer suggestion and used a t-SNE method (Maaten & Hinton, 2008) to visualize high dimensional dataset in a 2-dimensional space. This new analysis clearly shows that metabolism of *E. coli* populations evolved in the presence of the antibiotics move away from wild-type strain and populations evolved only in the presence of glucose or acetate. This effect is stronger in glucose minimal medium, where metabolic plasticity is larger. We now refer to this analysis in the main text. It is worth noting that similar conclusions were drawn more specifically for central carbon metabolism in glucose evolved populations (Fig. 3 and related supplementary figures).

Then in Figs 3 and 4, it's unclear which mutations are causal and which are passenger-this could be done to some approximation by screening the Keio library in these conditions or at least using the Nichols et al. data for this (with the caveat that this data is in LB).

The reviewer brings up an important point that we have also thought about a lot. We have tried to enrich for mutations likely to be directly related to antibiotic resistance by only considering the most frequently mutated genes among the four lineages. It is worth noting that while we observed significant heterogeneity in the overall landscape of mutations detected by genome sequencing for the lineages evolved under similar selective pressure, the same lineages seem to converge to a relatively more similar metabolic steady state, in particular on glucose minimal medium (Fig. 2A and new figures EV1-2).

We appreciate the suggestion to use the Nichols et al. data to distinguish functional mutations. In fact we take precisely this approach to corroborate our findings on the role of efflux pump expression mediating chloramphenicol resistance, with additional evidence provided by the extensive literature regarding the effect of mutations in the associated regulators (Swick *et al.*, 2011) (Suzuki *et al.*, 2014). However, for less well-characterized mutations, we feel that the approach is too superficial, due mainly to the differences in experimental conditions, but also to the fundamental difficulty of simply predicting whether a mutation causes gain or loss of function. We feel that a more rigorous approach, out of the scope of the current study, is required.

3) Explanation for evolution of CHL resistance during growth in glucose: this part has weaknesses.

First there are inconsistencies. Authors claim: "Since acetate metabolism depends on respiration, a similar compensatory mechanism to evolving chloramphenicol resistance was not feasible." Yet in Fig 3, *arcR* and *marA* have also mutations in acetate evolved strains. Authors should check if mutations on *acrB/marR/acrR* have an impact on CHL resistance during growth on Glc and Ace (btw is not clear in Fig S11 in what growth media the experiment is done- see also minor comment 14).

The reviewer brings up a good point. While it is true that mutations in *acrR* and *marA* also occur in acetate-evolved strains, their impact is less significant and does not lead to the rapid acquisition of resistance observed on glucose medium (Figure 1 shows the generally slower evolution of resistance on acetate). The reason is that acetate utilization is restricted to respiration for energy generation, while growth on glucose can proceed either

via respiration or fermentation. Thus, if overexpression of efflux pumps constrains respiration, cells growing on glucose are able to compensate much more readily than cells growing on acetate.

Experiment for generating Fig. S11 was performed in glucose minimal medium and is now explicitly stated in the figure legend.

Second, fermentative vs respiratory metabolism will impact drastically pmf generation, which is the energy source of the AcrAB-TolC pump. Could it be that mutations on negative regulators of the pump, boost up the levels of the pump, and metabolic reorganization in Glc, tunes in the activity of the pump - so it is active only as much as needed. Pumps are usually not very selective on what they pump out (so may be pumping out also necessary metabolites) and come with a high energy cost. There are simple experiments to check for this scenario: a) probe levels (western blots) and activity (Nile red assay) of AcrAB on Glc and Ace-evolved strains containing e.g. *marR* (or *acrR*) and compare it to corresponding Keio mutant (as non-evolved); b) compare contribution of *marR* or *acrR* mutations to CHL resistance in evolved vs non-evolved strains (it is easy to revert the evolved strain to *marR*⁺ or *acrR*⁺ with P1 transductions).

We did not focus on experiments to probe AcrAB expression, since the fact that loss of function mutations in *acrR* or *marR* increase expression of AcrAB is extremely well characterized and accepted in the literature (Suzuki *et al*, 2014). The reviewer does present an intriguing hypothesis for an alternative way in which evolution could compensate for the cost of this increased expression, by using metabolic rearrangements to adjust the concentrations of effectors of pump activity. We are certainly interested in this idea, but we think that the proposed experiments would not be conclusive – e.g. Nile red assays simply test for maximal activity in the presence of an unnatural substrate, while the proposed regulation must probably be substrate-selective (otherwise the benefit of higher chloramphenicol export is lost). In our opinion, more thorough experiments to test this hypothesis are beyond the scope of the present study.

We do want to comment on the nature of the cost of efflux pump overexpression. Firstly, if the major cost of efflux pumps was energetic, one would expect to find the opposite metabolic phenotype, where more glucose is respired in order to maximize ATP yield. Secondly, we investigated the hypothesis that the cost is due to increased secretion of necessary metabolites, but using extracellular metabolomics data. We could not find strong associations between accumulation of specific extracellular metabolites and mutations in efflux pumps, arguing against this hypothesis. A partial caveat is that the extraction methods in this work are biased towards small polar metabolites, while AcrAB substrates tend to be highly hydrophobic, meaning that further effort to investigate this hypothesis may be warranted. Nevertheless, the evidence points to proteome (particularly membrane proteome) reorganization as the major cost of efflux pump overexpression (Basan *et al*, 2015; Nichols *et al*, 2011; Zhuang *et al*, 2011), and idea that we are following up in separate work.

Third, evidence for connection postulated about membrane proteome reorganization and CHL is unclear. The authors state: "Circumstantial support for hypothesis ... comes from phenotypic profiling of the *E. coli* gene deletion library where many chloramphenicol resistant mutants were more sensitive to cell wall damaging agents or oxidative phosphorylation inhibitors". First, it's unclear how authors assess this. Do they look for anti-correlation between these conditions? Fig S12 and S13 do not help to understand how comparison is made (btw Fig S13 does even not exist, and legends for Figs S12 and S13 are the same). It is also non-intuitive why mutants that potentially have an increased fermentative metabolism (important for chloramphenicol resistance) are sensitized to oxidative phosphorylation inhibitors.

We made a mistake in referencing the figure and actually referred in the text to Fig. S16 of the supplementary.

In this analysis we correlated the vector of epistatic interactions associated to chloramphenicol against all other perturbing agents. Since multiple concentrations of the same perturbing agent were tested the box plot reports median and first/third quartiles of Spearman correlation against each perturbing agent/dosages. We observed a tendency for gene knockouts that aggravate the effect of chloramphenicol to buffer the effect of perturbing agents such as EGTA, ampicillin, amoxicillin or CCCP, while knockouts that buffer the action of chloramphenicol are deleterious for EGTA, ampicillin, amoxicillin or CCCP.

By rearranging the proteome resources to switch from respiratory to more fermentative metabolism, we move from a condition of high proteome cost (due to the expression of costly enzymes in TCA) to low energetic yield (Basan *et al*, 2015). Energy supply is likely to become more critical limiting factor upon impairment of ATP synthesis by CCCP action on the residual oxidative phosphorylation. We now include this discussion in the supplementary materials.

4) Norfloxacin results are only discussed in the last sentence of Results. There should be either more extensively presented (with main or Supp figure) and discussed or the authors should take these results out completely. Added value for including them in the manuscript, as it currently stands, is little to none.

We did not follow up on the gyrase inhibitors mainly because the difference across glucose and acetate evolved populations were more subtle and difficult to interpret. Consistently, the mutations acquired in acetate and glucose minimal media were more similar than for other antibiotics, such as chloramphenicol or ampicillin. Nevertheless, we thought that discussing this aspect even if briefly in the manuscript was important. We now moved this into the discussion section.

Minor comments

1) Abstract:

a) 2nd line: maybe "interaction" instead of "synergy"

OK

b) "both in terms of rate and mode of resistance acquisition": since most people will think "rate of resistance acquisition" is equal with "rate of mutation", maybe is better to rephrase or specify here.

OK

c) Last sentence: is unconnected with rest of text - it is not clear how this will help to fight antibiotic resistance (btw this is never discussed in any extent in main text - although it would be beneficial)

Understanding resistance and compensatory mechanisms could offer potential new strategies to slow down evolution of resistance, such as identification of new potential targets for multidrug therapies and potentially to individuate weaknesses in evolved resistant bacteria (such as fosfomycin cross sensitivity conferred by ampicillin/glucose) (Gonzales *et al*, 2015) (Lázár *et al*, 2013). We now have this point discussed in the conclusion of the main text.

2) Intro: "typically, such resistance mutations or genes come at a fitness cost that reduces rate of bacterial proliferation": I am not sure if you can make this as a general statement. It's an expectancy, but there is no systematic study addressing it - or if there is please cite it. For example, many of the pump mechanisms of resistance do not change growth rate.

We rephrased the text. However, while the cost of resistance is highly variable (Vogwill & MacLean, 2015) and some efflux pumps seem not to have an effect on growth, other, like AcrAB overexpression do negatively affect growth rate (Mingardon *et al*, 2015). As discussed earlier, we think that is in fact the basis of the metabolic rearrangements observed. A number of other studies have consistently described the fitness cost associated with antibiotic resistance, and we now cite these in the text.

3) Was pH monitored during the evolution experiments? 0.4% Glucose and respiro-fermentative growth will eventually lead to media acidification (which will depend on how media is buffered - but with M9 0.4% Glc, you are on the border of being able to buffer it). According to authors, the evolved strains in Glc seem to secrete even more acetate. Since pH has a dramatic role in the uptake/activity of antibiotics (e.g. decreases uptake/effectiveness of fluoroquinolones but increases that of chloramphenicol) the authors should check the pH (along the growth curve) for at least the final evolved strains.

pH was not monitored during the evolutionary experiments. While wild-type cells that grows up to stationary phase (OD of approximately 5) can acidify the pH from 7 to about ~5.5, it is worth noting that due to the presence of the antibiotics and selection of populations that survived to the highest dosages, we rarely propagate cells that reached an OD higher than 2. This also partially answer the subsequent point 4). While doubling time of wild-type *E.coli* in glucose is ~60 minutes and is ~180 in acetate, it is hard to estimate how much time these

evolved cells spent in stationary phase, since we monitored OD only at the end of the 48 growing cycle. As previously mentioned, given the growth reducing action of the antibiotic and the cost of accumulated mutations we expect to catch several of these evolved population in early to mid log phase, for the subsequent propagation in fresh media+antibiotic. Moreover, as mentioned by the reviewer, while changes in pH are expected to have completely opposite effect on the action of chloramphenicol and norfloxacin, the slower speed of resistance acquisition is observed in acetate minimal medium for both antibiotics.

4) What is the doubling time in Glc vs Ace? With a 48hrs cultivation cycle, it would be interesting to know how many hrs bacteria spent in stationary phase in each case. With stress (stationary phase)-induced mutagenesis playing a role in mutation rate, this could be a factor.

Please refer to the above point 3

5) Page 3: "slowly growing cells being more resistant to antibiotics": this is a registered knowledge for decades, so citing a paper from 2014 for this seems "inappropriate".

We changed the reference.

6) Page 4 & Figures 2A/B; Figures S1/S2; if the statement is that lineages evolved in same stress/media behave same, then it would be better to cluster the correlations of metabolic profiles and not put them in consecutive order.

This will be much more confusing picture to read. We now included some more pictures in the supplements (Fig. S1 and S2) to clarify this point and reinforce our statements. Please refer also to comment 6 of reviewer 2.

7) The definition of Evolved Metabolic Characteristics (EMCs) is scattered over the methods section, figure legend and main text. Furthermore, it is unclear what precise metabolites were used to calculate the SH for each reaction i (presumably the ones involved in reaction i). It is also unclear how the concept of "negative shadow prices" was included in this formula and whether absolute values were used or not.

We used a permutation test to assess whether given an objective function (e.g. maximization of flux in reaction X), limiting metabolites (i.e. metabolites with negative shadow prices) are overrepresented among metabolites with a measured change in relative abundance during evolution of resistance to an antibiotic (selected using the R2 values from sigmoidal fitting). For each tested objective function, and hence for each tested reaction in the model, all metabolites (including co-factors) are taken into account.

Moreover, the directionality of the metabolite changes (e.g accumulation/depletion) is not discriminative in such a framework. For example higher demand for a metabolite can induce its overproduction and accumulation, or an increased utilization, resulting in a decrease of metabolite levels. In both cases the metabolite can still be limiting. This is the reason why the sign of the measured metabolic changes was not taken into account to establish the link with shadow prices. The statistical procedure tests how likely it is that the overlap between predicted limiting metabolites for reaction X, and metabolites with an altered abundance upon resistance to antibiotic A, would be found by random metabolite selection.

To avoid introduction of too many technical details in the main text, we included the detailed description of the statistical analysis in the corresponding method section.

8) As a more general comment, referring to the methods section is missing throughout the manuscript. This would be very helpful for reader to know when he/she should read more to understand the underlying assumptions/work for this statement.

OK

9) Page 6: the authors state "Intermediates in some metabolic pathways affected upon direct antibiotic exposure in wild-type E. coli featured similar persistent changes in evolved resistant populations, often independent of the nutrient environment used during selection, such as glycerophospholipid and nucleotide metabolism in ampicillin and norfloxacin evolved strains, respectively (Fig. 2D)." The expression "some metabolic pathways" is ambiguous and it remains unclear what % of the EMCs are indeed not affected by antibiotic treatment. Furthermore, the reference for Fig. 2D after this sentence might make one think that the data shown in the figure

comes from antibiotic treated bacteria. I'm not sure whether that is the case, due to the "Acetate" and "Glucose" separation, but nothing is mentioned in the figure legend in this respect.

We rephrased this section. We make clear that the “immediate response” to antibiotics in wild-type E.coli was tested only on glucose minimal medium.

10) Page 6, next sentence: "The majority of ECMs, however, locates in metabolic pathways, which are not directly affected by the respective antibiotics". This implies that previous discussed pathways have to do with the mode of action of the two antibiotics, but to me this not obvious why.

The basic premise is that metabolic changes directly induced upon antibiotic treatment are somehow directly caused by inhibition of the drug target (e.g. gyrase upon norfloxacin treatment). On the other hand metabolome profiling of evolved resistant populations monitors the metabolic changes in an antibiotic-free medium. Hence these changes are not caused by the action of the antibiotic, but are the results of acquired mutations, which can directly impinge on drug targets or associate to other resistance/compensatory mechanisms. We clarified this point in the text.

11) Authors observe similar mutation rates at both carbon sources, and evolution experiments are carried for same number of generations. So expectancy is that evolved strains carry same number of mutations. Yet Amp-Ace evolved strains have 4x more mutations than Amp-Glc ones (Fig 4D), and both are quite different from Chl evolved strains (Fig 3C). Authors should comment on this.

One possibility is that even at similar mutations rates the probability of a mutation to be neutral is much higher in acetate minimal medium. It should be noted that we sequence evolved populations, not clones, so this effect will be quite significant. It is also quite possible that there is a larger genetic space of mutations that confer the same resistance phenotype in some conditions compared to others. We now comment on this in the discussion.

12) Page 7: "Indeed the evolved populations exhibited significantly higher rates of glucose consumption, acetate secretion and a reduced relative oxygen uptake ... (Fig. 3B and Table S5)". As no detailed information on the statistical test used to support this statement is provided, the usage of the term "significantly" can be misleading. Additionally, "Estimated fluxes" in the legend of Fig 3 is ambiguous. Estimated by model simulations or calculated from experimental measurements?

We rephrased the text and explicitly mentioned that all data in Fig. 3B are experimentally measured fluxes.

13) Please provide more detailed explanation on how the p-values in Table S7 were calculated. Did you correct for multiple testing?

Btw do these mutations co-occur in the same strains? Since some of them work through the same pathways, I cannot really see the added value of having both.

We first selected only genes with at least one mutation in at least three of the four lineages evolved on glucose. For each gene, the absolute difference between the average measured oxygen uptake rates in evolved populations with and without mutations was calculated (t_{obs}). Significance of the difference was estimated by means of a permutation test and sorting the estimated p-values. P-values were calculated as following: for each gene, the populations with and without mutations were randomly shuffled 1000 times. In every permutation the absolute difference of averaged oxygen uptake rate between the two groups (e.g. mutated vs non-mutated) was calculated (t_{rand}). P-values are calculated as the probability that random assignment of populations with and without a mutated gene yield a larger difference in oxygen uptake: $pvalue = \#(t_{perm} \geq t_{obs}) / 1000$.

We speculate that the benefit of bearing mutations in genes that are partially overlapping (on the same pathway or have a similar function) is a synergistic increase of their expression/activity levels, although at this stage we don't have data supporting this claim. It should also be noted again that these are sequenced populations, so it is possible that different subpopulations have mutations in different pathway gene

14) Fig S11: not clear how significant the *acrR* effect is. It is certainly not strong. Also clarify in main text that you are talking about wildtype *E. coli*, when describing this Figure.

The reviewer is correct. Deletion of *acrR* yields only a minor tolerance to chloramphenicol, and much stronger is the effect of $\Delta marR$. These results are consistent with results reported in Nichols et. al. It is possible that deletion alone of *acrR*, under the tested laboratory condition, is not sufficient to elicit an optimal overexpression of the efflux pumps. This seems also to be evident in Fig. S14, where acetate secretion is much more pronounced in a $\Delta marR$ strain. We now clarify that the experiment is performed on knockout strains of wild type *E. coli*.

15) Testing the Ace evolved mutants in Fosfomycin would be a good control to include.

We thank the reviewer for this suggestion. We have now included the suggested experiment and results in Supplementary Fig. S12. In agreement with our expectations, we found that ampicillin resistant populations evolved in acetate minimal medium have a similar or even higher tolerance to fosfomycin (opposite to glucose-ampicillin resistant populations). Surprisingly, most of the evolved population in acetate-ampicillin exhibited a growth advantage with relatively low fosfomycin concentrations, and one population in particular exhibited a strong benefit from the presence of fosfomycin. In this evolved population, growth-rate is 60% slower than wild-type *E. coli* in acetate minimal medium, and it increases with higher fosfomycin concentrations. We comment on this in the supplements but at the moment we can only speculate on whether this phenomenon can be related to the role and cost of cell wall recycling in the presence of a non-PTS carbon source.

16) The fact that Ace evolved strains in Amp have so many more mutations - many of which as authors point are in pleiotropic transcriptional regulators, but yet they show they least amount of metabolic rearrangement and resistance to Amp, illustrates very well: a) the confinements metabolism puts in phenotype evolution (main message of paper), but also b) the metabolic plasticity may actually buffer for the selective pressure to evolve (genetically mutate). Latter is only briefly discussed in the text and in my opinion warrants more space.

We thank the reviewer for this comment. We now include this point in the discussion.

17) Conclusions: "Emerging ..., Achilles heels if they are exposed to other pertrubations"; this a typical case of "collateral sensitivity" where a lot of work has been published lately and would be use to incorporate in discussion.

We now discuss this in more details in the discussion section.

Reviewer #2:

"Metabolic constraints on the evolution of antibiotic resistance," by Zampieri and colleagues investigates a very important question; how does nutritional environment impact the development of resistance to antibiotics. The authors performed adaptive evolution studies, quantified a large number of intra- and extra-cellular metabolites, and used several computational tools in order to sift through the mountain of data and generate hypotheses. Overall, this reviewer believes that the goals of the study are important and there is a lot of potential for this type of work; however, there are numerous concerns and questions that remain, which need to be addressed and these are differentiated as major and minor issues below.

Major.

1. Fig. 1: How were generations calculated? This is an incredibly important element of the evolution experiments that I do not see explained. Some of my below comments will convey its importance.

Number of generations during each passaging step were calculated by (i) measuring the final OD after a 48 hours growing cycle (OD_{fin}), (ii) 9 μ l of selected evolved populations were reinoculated in 900ul of fresh medium yielding a starting OD equal to $OD_{fin}/100$. At the end of the 48 hours

growing cycle OD is measured (OD*) and number of generation is calculated by the following formula: $\log_2(\text{OD}^*/(\text{OD}_{\text{fin}}/100))$. We clarified this in the materials and methods.

2. "...resistance to all three tested antibiotics evolved much faster on glucose than on acetate (Fig. 1C), demonstrating that environmental conditions can constrain the rate of resistance acquisition." First, Fig. 1C needs to be explained better in the legend (e.g., what are the blue and red shading from, and how do authors quantify generation).

We rephrased the legend and add the description in point 1.

Second, the authors say that resistance evolved faster on glucose than acetate, but I do not see the statistical test the authors applied to make this conclusion?

Qualitatively this is evident from the generation time it takes for cells in glucose vs acetate to grow in the same antibiotic concentration. Please see also the following point below.

Third, I find it a bit problematic to use a fixed time (48 hr) to transfer the cultures to the next antibiotic-containing plate for media that have vastly different growth-rates. This is compounded by the fact that transfers were made from wells that reached an OD of 0.5 regardless of the media source. What was the maximum density of the glucose-grown cells w/o antibiotic compared to that of the acetate-grown cells w/o antibiotic? Based on the amount of each used as described in the methods (5 g/L), I suspect that the total amount of growth the glucose could support is much higher than that of the acetate. So with using a fixed time and fixed OD cutoff threshold for transfer, I would expect that slower growth-rates and lower potential final densities would bias results (e.g., for acetate populations the wells used for transfer to next round would come from lower antibiotic concentration wells).

We thank the reviewer for raising this concern and giving us the chance to better explain the experimental setup. To compare the speed of evolution toward antibiotic resistance (Fig. 1C) we are always comparing the number of generations it takes for the cells (in glucose and acetate minimal media) to grow in the same antibiotic concentration. This comparison is independent from both growth rate and the maximum OD cells can reach in the two media. Indeed, time wise, evolving *E. coli* cells in acetate minimal medium for the same number of generations as in glucose, took us longer. Typically cultures in limiting levels of antibiotic reached an OD between 0.5 and 2 after 48h.

To be unbiased in this regard, I would think that the transfers should occur at a time dictated by the unstressed control (e.g., when they reach stationary phase) and transfer should occur at a density that corresponds to a fraction of the density obtained for the unstressed control (e.g., MIC50, or concentration where 50% of growth has been inhibited). I think the authors should adjust their conclusions based on these confounds, or acknowledge them and explain why they do not think they matter with regard to their conclusions.

This strategy while certainly valid, adds several complications. Throughout evolution, resistant strains exhibited different physiology (e.g growth rates) even without the presence of the antibiotics (as shown in this study), so it would be unclear which strain to take as a reference to test for the time it takes to reach stationary phase. Moreover, we don't know a priori which is the maximum concentration of the drug the evolved strains are capable of growing in. Our experimental setup was design to always systematically test at each passaging step a range of different antibiotic concentrations, so that we could monitor how fast evolved strain under different nutritional environment can "climb" the antibiotic gradient (Fig. 1B). While we agree with the reviewer that in principle transfer of cells that are in an identical growth phase would be ideal, this would have been practically very difficult to achieve, and not crucial for the conclusion drawn in this study.

The only practical way to actually implement reviewer suggestions would have been to use the approach developed by Roy Kishony and colleagues (Toprak *et al*, 2012)(i.e. morbidostat) in order to fix the same growth rate of evolving *E.coli* in glucose and acetate minimal medium. However, even this setup has some limitations, and one could question why and how a specific dilution rate was selected. More important, and as also mentioned by the reviewer, having different growth rates in the two nutritional environments can play a role in imposing different metabolic "constraints".

Overall we feel that this is a very fruitful discussion, but are convinced that our inoculation strategy is not unreasonable and that the main conclusions are not affected by it.

3. Fig 2a: This is data from ~190 different populations from different times in the trajectory of the 24 evolved lineages with controls. This is from populations correct? Not individual colonies from plating of those populations? So it is possible that the data reflects the metabolic characteristics of a population and not of individual resistant mutants?

Correct, we always refer to populations in the text. We profiled the metabolome of the same populations we passaged at the end of the 48 hours growing cycle. We clarify this in the text.

4. Fig. 2c: y-axis says significantly changed metabolites. What statistical test was used? It read as if these were just the metabolites with a good Rsquared value for the sigmoidal fits. How does that assesses significance?

Metabolic traits that are the results of fixed mutations in evolved resistance mutants, and not the mere results of spurious changes or neutral mutations affecting metabolic processes, are expected to show a robust and sustained transition from wild-type basal levels to new (different) steady state levels. We discriminated transient from permanent metabolic changes by fitting a sigmoidal curve on the metabolite profiles observed in each lineage during evolution of antibiotic resistance (i.e. metabolite evolutionary trajectories such as pantothenate in Fig. 2). A sigmoidal model intrinsically describes and captures slow and rapid changes from one basal state to a new different state. R-squared is a statistical measure of how close the data are to the fitted regression line and reflect the goodness of fit. For those metabolites where a sigmoidal curve exhibits poor descriptive ability (low adjusted R^2) we assumed either:

- Data are too noisy
- There is no significant change in the relative abundance of the metabolite during evolution
- Changes are transient and reabsorbed to a normal basal level by the end of our evolutionary experiment, presumably because they are non-adaptive changes. Hence our sigmoidal model is not able to describe the data and will yield a poor R-squared.

We rephrased the text also in the figure legend to make sure that reported metabolites are selected based on their R-squared value above an arbitrary high threshold of 0.6.

5. "Indeed the evolved populations exhibited significantly higher rates of glucose consumption, acetate secretion and a reduced relative oxygen uptake, revealing a switch from respiratory to fermentative metabolism (Fig. 3B and Table S5)." So these were populations right, not colonies obtained from plating of populations that grew at specific antibiotic concentrations? Important for next point.

Correct. These were the populations at the end of the evolutionary experiment, and physiology data comes from growth in glucose minimal medium without any antibiotics.

6. "To this end, we performed genome sequence analysis of the 26 strains at the end point of evolution, and used multivariate statistical analysis to identify putative mutations whose presence correlated with measured respiration rates in glucose evolved mutants (Table S6)." Were these 26 colonies or 26 populations? It matters immensely for interpretation of the whole genome sequencing results. In fact, can the authors please be more explicit in their methods as to how they "called" a mutation. It seems that they required there to be a 20X or higher coverage, but what constituted a mutation, what allele frequency did you use? Its important to know this as well as whether colonies or populations were sequenced.

We rephrased the text so to make clear that we investigated only populations and not single colonies. The mutations have been reported with the default MAF of 1 %, which means that in each samples the alternative allele had to be observed at least 1 % of the total and it would be called heterozygous, whereas a minimum 80 % freq would be required for a snp to be called alternate homozygous.

7. "Consistently, we showed that deletion of the efflux pump repressors MarR or AcrR (Nichols et al, 2011) not only rendered *E. coli* more tolerant to chloramphenicol (Fig. S11-13), but also caused a strong increase in glucose fermentation via acetate secretion (Fig. S14)." This is an interesting result, but it is not the same as repairing the promoter mutation in the evolved strain, and analyzing its metabolism and resistance. To repair the mutations in the evolved strain and perform these experiments would establish causality. The data with the repressor deletions does not convince me of much with the evolved strains. Also, you could take the parent strain and use the same promoter mutation your sequencing data found. That would be much better than the repressor deletions. Experiments with evolved strain assess if the mutation is necessary, experiments using the promoter mutation in the ancestral strain assess if the mutation is sufficient.

We agree with the reviewer. However, the main messages from our study are that (i) metabolome undergoes large rearrangements during evolution of antibiotic resistance and offer an independent and complementary view to the analysis of genetic mutations that are necessary or sufficient to yield resistance. (ii) such metabolic changes can be functionally interpreted by a new approach capable to predict flux rearrangements and to suggest less obvious compensatory mechanisms. We think following up on the role of individual single point mutations is out the scope of this study and we now discuss this more openly in the discussion.

8. "In support of this hypothesis we demonstrated ampicillin-glucose evolved mutants to be hypersensitive to fosfomycin (FOS) (Fig. 4C), an inhibitor of the last enzymatic step of the anhydromuropeptide recycling pathway (Fig. 4B)." The authors neglected to mention that the target of FOS (*murA*) is an essential enzyme in peptidoglycan biosynthesis, not just the recycling pathway.

We clarified this in the text.

Also, was this done with populations of mutants or individual mutants that had come from colonies on a plate? The authors should devise other means to test the importance of recycling independent of peptidoglycan biosynthesis (perhaps something with *ampD* or repairing something in the mutants that is responsible for the peptidoglycan recycling).

All experiments are performed on populations. As in the preceding example our main goal was not to find the exact mutations responsible for compensatory mechanisms, but rather suggest and explore the condition dependency of such potential compensatory mechanisms that involve metabolism.

9. "Differential propagation of resistance and compensatory mutations under different nutrition conditions (Fig. 3C, 4D) reflect the potential role of metabolism in rendering certain mutations less accessible by natural selection." First, why are 3C and 4D generated by having a different threshold of mutations, 2 out of 4 for 3C and 3 out of 4 for 4D? Second, since only 4 populations were propagated for each media and antibiotic combination, how can the authors make this claim? If the mutation is less accessible, the authors should see a statistically significant difference in mutations to specific loci in glucose plus antibiotic when compared to acetate plus antibiotic. Was this statistical comparison made to identify loci (e.g., *murE*, *rpoD*, etc.) that are statistically less accessible between conditions?

The threshold was adapted to find the most recurrent mutations under the two different selective pressures (i.e. antibiotics across the four lineages) and have still readable image. Overall we ended up comparing 8 lineages evolved under the selective pressure of the same antibiotic; 4 in glucose and 4 in acetate minimal media. We agree that this number is probably not large enough to draw indisputable conclusions. We try to discuss this limitation explicitly.

Minor

1. First sentence of the summary, "While we begin to understand the genetic basis of antibiotic resistance, ...". This is highly inaccurate. Research on antibiotic resistance has been conducted for

over 60 years, and the genetic basis for resistance is understood in a vast majority of cases. Antibiotic resistance forms the basis of a lot of micro- and molecular biology, and writing of that field in this way will turn off many people. I would advise adjusting this statement.

We do agree with the reviewer, and in fact our intention was to acknowledge the tremendous amount of work in the field. Nevertheless, we are also aware that our current understanding on the landscape of potential beneficial and compensatory mutations is far from comprehensive. Anyway we rephrased the sentence accordingly.

2. "A constrain-based modeling approach to...": should be "constraint".

OK

3. Fig. 3b: "Glycolisis" should be "Glycolysis".

OK

4. Fig. 3b: Can the authors please describe what all of the data points and box-plots are here, I don't find the legend very helpful in this regard.

OK

5. 5. Top of pg 9: rpoD is a sigma factor, not a transcription factor.

OK

Reviewer #3:

The present manuscript explores the important question of metabolic constraints on the evolution of antibiotic resistance. The experimental design is clearly described, but unfortunately, lack of clarity in presentation of the results and shadow price method makes it hard to evaluate the extent of progress on this important problem.

1. Metabolomics relies on direct ionization without LC separation. While this is a very interesting method and can be used to achieve things that cannot be done with regular metabolomics, it is no substitute for LC-MS due to numerous interferences due to in-source degradation, isomers, etc. It seems likely that the number of reported measured metabolites is greater than the number actually measured (indeed, it seems that the authors count single measured ion as if they measured multiple isomeric metabolites) and more importantly that many are mismeasured due to unknown interferences.

Indeed the employed method injects samples directly in the mass spec without any chromatography. This method is extensively described in (Fuhrer *et al*, 2011), and has been now used in far more than 20 applications (Link *et al*, 2015; Schulz *et al*, 2014; Sévin & Sauer, 2014; Zimmermann *et al*, 2015), (Schulz *et al*, 2014) (Mirtschink *et al*, 2015; Gomez de Agüero *et al*, 2016; Zampieri & Sauer, 2016) yielding important insights. We are well aware of its limitation over chromatography-based methods, namely the lack of separation of compounds with similar m/z, difficulty of detecting in-source fragments, the risk of matrix effects. The purpose of the method is to be able to handle large data sets for exploration and rapid hypothesis generation. The primary focus of this work here is precisely this: hypothesis generation from large data sets.

We'd like to stress that the aforementioned issues are taken into account in our computational analysis. We do not rely on a single or few measurements but essentially the entire set of putatively annotated metabolites. The confidence in the analysis comes from the overall consistency and not from a potential mis- or ambiguous annotation of an ion peak. We do not make any specific assumptions on ambiguous annotations of metabolites with identical masses, but simply include all possibilities to derive robust predictions on their functional meaning in a network context. This ambiguity is reported in the method section and was clarified in the main text. One key contribution of our work is precisely to cope with this ambiguity by developing a computational framework that allows deriving pathway flux hypotheses from nontargeted metabolite data. Thus, the modelling was

not only essential in enabling us to identify metabolic pathway adaptations, but also in itself a contribution that we expect to be useful to many colleagues that work with metabolite data, and specifically nontargeted metabolomics.

Regarding the so called “matrix effect”, this is largely set by the organism and condition. Since we always extract *E. coli* cells with the same extraction buffer, and normalize data within similar nutritional condition, we can safely assume the matrix composition of our samples to be largely constant across samples. Furthermore, for several metabolite, we could prove that intensities scales linearly with the corresponding metabolite abundance, and we now included in the supplements a new set of experiment where we related the actual concentrations of 64 different compounds to the intensity measured by our direct injection method (Fig. S14). For the vast majority of compounds, measured intensities scales linearly with actual concentrations in a large range. This analysis doesn't account for the occurrence of unknown contaminations or in-source derivatives, but these were rarely found to be predominant in extensive correlations analysis performed in large-scale datasets (including an *E. coli* study with ~4'000 mutants under similar experimental conditions). Since the risk of mis-measurements remain, rather than trying to resolve the exact magnitude of the metabolite abundance in our computational analysis we rely on relative metabolic changes (see point 2 for additional consideration on this point).

Even though quantitative and targeted metabolomics analyses aren't a necessary precondition, we agree with the reviewer that this type of data in principle can offer an easier interpretation. However using classical targeted LC-MS approach would have been prohibitive due to the large sample size and to the relatively low coverage. Overall, our novel metabolomics-based investigation strategy represents an unprecedented methodology to study evolution of resistance to antibiotics that can be potentially extended to study evolution of resistance to other drugs or biological systems. We now discuss the limitation of this approach with respect to a more targeted and quantitative mass-spectrometry approach in the discussion.

2. The way of detecting metabolite changes relies very much on the "smoothness" of the evolutionary trajectory, without taking into account the quantitative extent of the change. Indeed, the quantitative extent of change in metabolite to show nowhere in the main text. It is hard to evaluate the paper without better sense of what the actual data look like.

It is precisely due to the technical issues mentioned in the reviewer's first comment above that we do not rely extensively on the exact magnitude of the metabolite abundance changes, as these may scale non-linearly with the actual metabolite concentrations. We consider the untargeted metabolomics data to be intrinsically semi-quantitative, and thus an additional dimension is needed to assess the significance of the intensity changes. Here we use the change in metabolite intensity over evolutionary time, which has the additional effect of filtering out sporadic metabolic changes that happen only at intermediate stages of evolution but are not retained in subsequent evolved population lineages. It should be noted that “smoothness” is not required – we allow the transition to be smooth over time or to have a step-like behavior. Both behaviors are captured by the fitting model.

The main value of the untargeted metabolomics method is to generate a large library of hypotheses regarding potential metabolic changes between strains or conditions. To give a better feeling of how the data look like we show the pantothenate example in figure 2 of the main text.

3. For reasons that are not well explained, the authors look for enrichment of negative shadow prices. As far as I can understand (although I cannot find the definition of the sign anywhere in the paper), in typical use of shadow prices, if one increases a metabolite like PEP, and you get more feasible flux through a downstream reaction like pyruvate kinase, the sign of the shadow price is positive not negative? Why the focus on negative shadow prices? It would be enormously beneficial for the authors to walk the reader through one or more real examples in the main text from data to shadow prizes to conclusions.

Shadow prices are calculated by assuming that a resource/metabolite is degraded by a „fake-sink reaction“. In the example provided by the reviewer (increase production rate of PEP lead to increase flux through pyruvate kinase) the shadow price is indeed negative. A positive shadow price would biologically mean that the metabolite (e.g. PEP) is not a limiting resource for the objective reaction, but rather a toxic element. In fact by sequestering the metabolite, cells would be able to increase the objective flux. Such examples are anyway very rare.

It is worth noting that shadow prices have a very precise meaning and are an established concept in linear optimization (widely used in economic problems) to identify “limiting resources” (negative signs): in our case limiting metabolites for a specific metabolic reaction under specific environmental conditions. We focus here only on the negative signed shadow prices mostly because we are interested in the concept of limiting resources and how these resources can constrain/shape evolution of metabolism in antibiotic resistant *E. coli*. We now clarify this aspect in the method section.

4. I do not know why the authors put much of the critical data, like the response of glycolysis to changes in MarR and AcrR in the supplement?

We have now moved some of these data/plots as Expanded View Figures

5. The extent of hypersensitivity to FOS is disappointingly modest. FOS concentration is on a linear axis and one strain seems to show only a < 2-fold change? I am not an expert in this area, but it seems like clinical interest would rely on order of magnitude changes?

We agree with the reviewer that the cross sensitivity to FOS is for most evolved populations in a relatively small range and we rephrased the text to avoid any overstatement.

6. In general, I find figure 2 fails to convey the most important information. Figure 2 should give a good sense of what is metabolically going on, in terms of metabolite concentrations and any measured uptake or excretion fluxes. Instead, we are left with correlations and overprocessed data that is still in the form of hard to comprehend lists. Much preferred to give a better sense of the actual concentration and flux data and then to pull out a few interesting observations to highlight (perhaps ones that relate to the larger shadow price approach?)

It is important to notice that there is very little processing of the data in Fig. 2. The relative concentrations among evolved populations (Z-scores) are directly used for pairwise correlation analysis (Spearman correlation). Please refer also to comments in point 2. We assessed the similarity in the overall metabolome rearrangements across evolved populations so to have a global and intuitive view on (i) the degree of drug specific metabolic changes, (ii) the influence of the growth media (iii) reproducibility of changes across the 4 lineages (iii) number of generations to transition from wild-type basal level to new steady state concentrations. We feel like this an optimal way to represent the high information content, and that it would be extremely complex and convoluted to give a good representation of the overall metabolome changes for more than 500 metabolites and 200 evolved populations.

The main message of this figure is to assess the extent to which metabolome in evolved populations differentiate between ancestor strains and different selective pressures. The examples the reviewer is mentioning are indeed given in Fig. 3 and 4.

7. Similarly, I find Fig 3 and 4 to convey little. The lists of evolved functions are divorced from any sense of the underlying raw data changes and yet are also very hard to read or understand. Figure 3B is much harder to read than normal bar graphs, and it is hard to match the colors to the conditions. Figure 3C/4D seem to me a very poor way to show the mutation information. Normally these type of diagrams are used to show the co-occurrence of different mutations (where every gene can connect every other gene, and the presence or absence of such linkages is informative). Here it seems that it could be much easier to read a table of mutations divided into acetate, glucose, and

both? This might also allow some chance to explain the nature of the underlying mutations and the function of the genes?

The EMCs listed in figure 3A and 4A constitute the premise of our follow up experiments. We think it is very important they are represented and we now try to explain these figures better in the legend. We also adapted figure 3B to improve readability. Concerning figure 3C and 4D, these are bipartite graphs that are indeed meant to convey the information of overlap mentioned by the reviewer. Genes that exhibit mutations under both conditions are double colored and link to both selective pressures. The list of genes the reviewer is asking is part of the supplementary materials.

8. The supplementary methods describing the shadow price calculations (and formulation as a symmetric dual problem) were incomprehensible to me. Better explanation is needed.

We further clarified the concept of shadow prices and their relationship with availability of metabolic resources in the main text and in the method section (see above). The underlying principle is that shadow prices are capable of individuating which resources (in our case metabolites) can be limiting for the cell to optimize an objective function, which in our case we considered to be maximization of flux through a metabolic reaction. Assuming that a resistant strain of E.coli is attempting to foster (optimize) more flux through reaction "X" we can use the shadow price calculation to systematically find which metabolites could be limiting for cells to achieve their objective.

9. To end on something positive, Fig 1 is lovely.

We thank the reviewer for the positive comment.

Overall, I do think that there is a lot of interest here, if it could be presented much better.

2nd Editorial Decision

28 August 2016

Thank you for submitting your revised study. We have now heard back from the three referees who were asked to evaluate your manuscript. As you will see below, while reviewer #1 is positive, reviewers #2 and #3 are not supportive and they raise substantial concerns, which preclude the publication of the study in its current form.

In particular, reviewer #2 points out that further experimental analyses are required to demonstrate a causal link between mutations (and metabolic phenotypes) and antibiotic resistance and mentions that the statistical significance of several of the reported findings has not been assessed. These comments are in line with the evaluation of reviewer #3, who was also not convinced that the changes performed during the revision process were sufficient.

As you may know, our editorial policy is to allow a single round of major revision. However, the reviewers note in their reports that examining how metabolic constraints affect the evolution of antibiotic resistance is a timely topic and that the presented approach and dataset is potentially interesting. According to our 'pre-decision cross-commenting' policy, we have circulated the reports and consulted with the reviewers on whether we could consider the study further if the remaining issues would be fully addressed. In reply, reviewer #2 emphasized that this would involve convincingly assessing the causal link of mutations and resistance (points 7, 8 of his/her report) and rigorously addressing his/her concerns on statistical significance. Moreover, we discussed with reviewer #1 over the phone and in his/her opinion addressing the points raised by reviewer #2 (including points 7 and 8 that involve additional experiments) seems feasible within the scope of a major revision.

Taken together, we would like to offer you a chance to address the remaining issues in an exceptional additional round of major revision. Of course, as you can probably understand, at this stage we cannot guarantee that the outcome of the review process will be positive. Acceptance of the

study will depend on convincingly addressing all the remaining issues raised by reviewer #1 and reviewer #2, including those mentioned above.

REFeree REPORTS

Reviewer #1:

The authors have put a lot of effort on revising the manuscript to address the reviewers' comments and to make it more easy-to-follow by the reader. This is a much improved version, with new figures, more complete legends and more conclusive text. Overall the points I had raised have been well addressed. I have only minor comments that have arisen from the rebuttal and reanalysis/rewriting.

1. There could be something less wordy conveying the same message as the first sentence of the abstract: "Despite our continuous improvement in understanding antibiotic resistance, how the environment impacts it remains unclear."
2. Why is the diagonal of heat maps in panels A and B of Fig 2 not red? This is comparing sample with itself so Spearman correlation should be 1.
3. Fig 1C: I did not think authors should change the scale from log to linear, but instead of having the power of 2 in the y axis, put the actual number so reader does not have to calculate what 2^{15} or 2^{18} is.
4. Explanation of evolution of CHL resistance:
 - a) Authors explain the fact that both Ace and Glc populations acquire Acr pump mutations, but only Glc populations gain by them by "Thus, if overexpression of efflux pumps constrains respiration, cells growing on glucose are able to compensate much more readily than cells growing on acetate". I agree this is plausible but also testable. The authors should also test then the *acrR/marR* mutants in acetate minimal media, and compare with Fig S7 - according to the hypothesis, *acrR/marR* mutations should help less if at all the CHL resistance in this case.
 - b) What I meant as an alternative model is that upon pump overexpression, the cell can still tune the activity of the pumps by going to a more fermentative growth, because this will lead to less pmf available for running the pumps (not that cell changes metabolism, to change the available pump substrates). This could be tested by Nile red assays, but I agree with authors it is more suitable to explore further in a future study.
 - c) Line of thought that connects membrane proteome reorganization and CHL is still weak, and I would change the way it is presented as a hypothesis the authors want to follow up in the future. I cannot see why membrane reorganization will make you more resistant to cell wall agents (target peptidoglycan in periplasm, and do not have to cross the inner membrane), and EGTA affects mostly outer membrane function. The effect of CCCP on fermentative growth, where you actually use ATP to create proton gradient is still confusing to me, but I can see the point of the authors if the cells are still doing some respiration which is important for them.
5. It is apparent that the Ace-AMP evolved populations do not depart much from the Ace alone population in terms of metabolites (EV2 - this not acknowledged in manuscript), which sinks well with the lower/slower AMP resistance (Fig1). At the same time the Ace-AMP evolved populations has 4x more mutations than the Glc-AMP ones (Fig 4D)! It looks as if in the Glc population a few mutations offering strong selective advantage (same that rewire metabolic capacity) are fixed in the different populations, whereas in the Ace population such jumps in the fitness landscape are not possible (due to the difficulty in rewiring metabolism) and thus cells explore more and milder solutions.
6. Although I get the reasoning on why authors feel safe that pH is not playing a role in their interpretation of the results, I would still like to see them measuring the extracellular pH of the

CHL-Glc evolved strains, when growing in minimal Glc media (these cells seem to be in a strong fermentative state). When *E. coli* aerobically ferments, it can acidify the external media to a degree that the reason it stops growing is this, and not because nutrients are exhausted (so it acidifies it in exponential phase). This depends on the amount of Glc available and the degree it switches to aerobic fermentation. If those populations are acidifying the media this can be a reason for not developing further resistance in CHL.

7. When comparing result of Fig S12 in the text, you could start with "In contrast" instead of "Differently".

Reviewer #2:

The authors' revision did not satisfy all of my concerns. Overall a lot of interesting phenotypic data is provided; however, the statistical significance of many results has not been assessed, which makes one question whether any differences are actually present, and the analysis of the results fell short of drawing causal links, which limits the amount learned from this study.

1. Fig 1C: The authors need to show statistical significance to assert that glucose-grown populations evolve resistance faster than acetate-grown populations. As it stands, this reviewer is not certain that statistical significance will be achieved with the data presented. The y-axis for 1C is odd as well, with it being log₂ transformed. Why can't the authors use a log-linear plot instead?

2. Fig. 2AB: For the colorbar, if it is depicting the spearman correlation, why is its value not uniformly 1 across the diagonal? It looks to be -0.2?

3. Fig. 2D: For the y-axis the word "significant" needs to be removed unless a statistical test was performed and reported.

4. "Indeed the evolved populations exhibited significantly higher rates of glucose consumption, acetate secretion and a reduced relative oxygen uptake, revealing a switch from respiratory to fermentative metabolism (Fig. 3B)": Significantly higher than what? What significance test was used? As it stands, this reviewer is not certain that statistical significance will be achieved with the data presented. Figure legend speaks of only CHL-glucose adapted populations, but much more data is presented than that, correct?

5. Fig. EV3: y-axis is odd. Plotted as %inhibition; 20% inhibited would grow faster than 40% inhibited, which I do not think is what the authors intended. Also, as it stands, this reviewer is not certain that statistical significance will be achieved with the data presented.

6. "Consistently, we showed that deletion of the efflux pump repressors MarR or AcrR (Nichols et al, 2011) not only rendered *E. coli* more tolerant to chloramphenicol (Appendix Fig. S7-9), but also caused a strong increase in glucose fermentation via acetate secretion (Fig. EV4)." This reviewer is concerned that the differences in S7 are not significantly different, and the differences in EV4 look like some could be significant but not others.

7. AcrR and MarR deletion mutants are not substitutes for working with the *acrB* promoter mutation you found or *acrR* mutation. Data from the deletion mutants can be used in a supportive fashion, but not for drawing causal links about the populations you evolved. Without performing the experiments I mentioned previously with respect to whether *acrB* promoter mutation is necessary and sufficient (or mutation in *acrR*), causality is not established, and without causality little is learned about the antibiotic resistance that was evolved in your experiments.

8. "Our EMC predictions (Fig. 4A) based on metabolite changes in resistant strains suggested recycling of anhydromuropeptides to play an important role in mediating resistance to ampicillin." Sensitivity to fosfomycin does not convincingly show that recycling of anhydromuropeptides is important to mediating ampicillin resistance. Altering the level of anhydromuropeptides and then observing increases/decrease in ampicillin resistance would be convincing. Further, doing that in your evolved populations would establish causality.

9. " $\log_2(\text{OD}^* - (\text{OD}_{\text{fin}}/100))$ ": this is incorrect. Should not be a subtraction but a division to capture generations. I presume this is just a typo.

Reviewer #3:

The present revision show some modest improvements without substantially addressing limitations in the data, analysis and figures. The paper does demonstrate that antibiotic resistance develops faster on glucose than acetate (an interesting result), and that there are antibiotic and carbon-source specific metabolic adaptations. As far as I can tell, it does not successfully validate recurrent antibiotic-resistance driving mutations, nor does it clearly establish strong metabolic phenotypes associated with resistance to specific antibiotics, beyond a modest trend towards decreased respiratory metabolism with chloramphenicol resistance, which is pretty logical given that chloramphenicol blocks protein synthesis which is a major ATP consumer. The new method with shadow prices is more clear now, but I still do not find its output to be particularly persuasive. The authors do not validate any strengths of this new analysis approach. Finally, the authors continue to claim measurement of over 500 metabolites in the abstract, even though the text has now been corrected to reflect their actual methods. More importantly, they continue to rely on the ampicillin-Fos example in the abstract, even though they admit that the finding is weak. Overall, my impression is that this is a timely undertaking, but did not yet yield the type of substantive discoveries that one might have hoped for.

2nd Revision - authors' response

18 December 2016

Reviewer #1:

The authors have put a lot of effort on revising the manuscript to address the reviewers' comments and to make it more easy-to-follow by the reader. This is a much improved version, with new figures, more complete legends and more conclusive text. Overall the points I had raised have been well addressed. I have only minor comments that have arisen from the rebuttal and reanalysis/rewriting.

We thank the reviewer for the positive and constructive comments.

1. There could be something less wordy conveying the same message as the first sentence of the abstract: "Despite our continuous improvement in understanding antibiotic resistance, how the environment impacts it remains unclear."

We rephrased it

2. Why is the diagonal of heat maps in panels A and B of Fig 2 not red? This is comparing sample with itself so Spearman correlation should be 1.

We thank the reviewer for pointing this out. There was a mistake on the colorbar. To scale the color codes between maximum and minimum Spearman correlation values across pairs of different evolved populations, we set diagonal values to 0 (rather than 1). We corrected the mistake.

3. Fig 1C: I did not think authors should change the scale from log to linear, but instead of having the power of 2 in the y axis, put the actual number so reader does not have to calculate what 2^{15} or 2^{18} is.

We did change the values on the y-axis according to reviewer suggestion

4. Explanation of evolution of CHL resistance:

a) Authors explain the fact that both Ace and Glc populations acquire Acr pump mutations, but only Glc populations gain by them by "Thus, if overexpression of efflux pumps constrains respiration,

cells growing on glucose are able to compensate much more readily than cells growing on acetate". I agree this is plausible but also testable. The authors should also test then the *acrR/marR* mutants in acetate minimal media, and compare with Fig S7 - according to the hypothesis, *acrR/marR* mutations should help less if at all the CHL resistance in this case.

The reviewer raise an important aspect that we now clarify. Expression of efflux pumps like *AcrAB* are certainly beneficial also in acetate minimal medium. Nevertheless we observed that compensation of their sustained overexpression in evolved strains on glucose minimal medium is associated with a rewiring of central carbon metabolism, involving higher fermentative glycolysis. Hence, we don't necessarily expect the efflux pumps to be less effective in glucose vs acetate minimal media, and therefore refrain from a non conclusive experiment.

b) What I meant as an alternative model is that upon pump overexpression, the cell can still tune the activity of the pumps by going to a more fermentative growth, because this will lead to less pmf available for running the pumps (not that cell changes metabolism, to change the available pump substrates). This could be tested by Nile red assays, but I agree with authors it is more suitable to explore further in a future study.

We appreciate the reviewer suggestions, and we agree this is an intriguing hypothesis.

c) Line of thought that connects membrane proteome reorganization and CHL is still weak, and I would change the way it is presented as a hypothesis the authors want to follow up in the future. I cannot see why membrane reorganization will make you more resistant to cell wall agents (target peptidoglycan in periplasm, and do not have to cross the inner membrane), and EGTA affects mostly outer membrane function. The effect of CCCP on fermentative growth, where you actually use ATP to create proton gradient is still confusing to me, but I can see the point of the authors if the cells are still doing some respiration which is important for them.

To avoid any overstatement we now rephrased the text and follow the reviewer suggestion

5. It is apparent that the Ace-AMP evolved populations do not depart much from the Ace alone population in terms of metabolites (EV2 - this not acknowledged in manuscript), which sinks well with the lower/slower AMP resistance (Fig1). At the same time the Ace-Amp evolved populations has 4x more mutations than the Glc-AMP ones (Fig 4D)! It looks as if in the Glc population a few mutations offering strong selective advantage (same that rewire metabolic capacity) are fixed in the different populations, whereas in the Ace population such jumps in the fitness landscape are not possible (due to the difficulty in rewiring metabolism) and thus cells explore more and milder solutions.

This is a possible interpretation. We explicitly discuss this point in the main text.

6. Although I get the reasoning on why authors feel safe that pH is not playing a role in their interpretation of the results, I would still like to see them measuring the extracellular pH of the CHL-Glc evolved strains, when growing in minimal Glc media (these cells seem to be in a strong fermentative state). When *E. coli* aerobically ferments, it can acidify the external media to a degree that the reason it stops growing is this, and not because nutrients are exhausted (so it acidifies it in exponential phase). This depends on the amount of Glc available and the degree it switches to aerobic fermentation. If those populations are acidifying the media this can be a reason for not developing further resistance in CHL.

The higher acetate secretion of CHL resistant strains with respect to wildtype will certainly have an impact on extracellular PH. However, these populations grew sensibly slower than wild type and it is very unlikely that during the evolutionary experiment we reached the maximum OD or that changes in PH caused by increased acetate secretion had a significant impact on any of our conclusions.

7. When comparing result of Fig S12 in the text, you could start with "In contrast" instead of "Differently".

OK

Reviewer #2:

The authors' revision did not satisfy all of my concerns. Overall a lot of interesting phenotypic data is provided; however, the statistical significance of many results has not been assessed, which makes one question whether any differences are actually present, and the analysis of the results fell short of drawing causal links, which limits the amount learned from this study.

We thank the reviewer for his technical comments which helped us to improve data presentation. We have done extensive additional statistical analyses as suggested, which did not change any of the main conclusions. Hence, we believe that we were able to address all statistical concerns, and also added new experimental evidence to reinforce our conclusions on the mechanisms underlying the higher fermentative metabolism in CHL resistant populations.

1. Fig 1C: The authors need to show statistical significance to assert that glucose-grown populations evolve resistance faster than acetate-grown populations. As it stands, this reviewer is not certain that statistical significance will be achieved with the data presented. The y-axis for 1C is odd as well, with it being log2 transformed. Why can't the authors use a log-linear plot instead?

We now revised the figure axis. To support the qualitative trends in Fig. 1C we now performed a statistical test in which we fitted the trends with an exponential function in which the drug concentrations are described as an exponential function of the number of generations:

$$D(g) = Ae^{\phi g}$$

Where D is the drug concentration at each passaging step, g is the number of generations and ϕ and A are the fitted parameters. As reported in the new figure EV1 we could show that for all three antibiotics resistance evolved significantly (pvalue<0.05) faster in glucose than in acetate minimal media.

2. Fig. 2AB: For the colorbar, if it is depicting the spearman correlation, why is its value not uniformly 1 across the diagonal? It looks to be -0.2?

We made a mistake in reporting the colorbar scaling values. Please see also point 1 of reviewer 1.

3. Fig. 2D: For the y-axis the word "significant" needs to be removed unless a statistical test was performed and reported.

To avoid confusion we change figure y-label.

4. "Indeed the evolved populations exhibited significantly higher rates of glucose consumption, acetate secretion and a reduced relative oxygen uptake, revealing a switch from respiratory to fermentative metabolism (Fig. 3B)": Significantly higher than what? What significance test was used? As it stands, this reviewer is not certain that statistical significance will be achieved with the data presented. Figure legend speaks of only CHL-glucose adapted populations, but much more data is presented than that, correct?

We now rephrased the legend of figure 3B to clarify that the data reported are for all the most evolved populations in glucose minimal medium together with the population evolved in acetate minimal medium without antibiotics. The differences we observed are highly significant as wild type glucose uptake and acetate secretion ~8mmol/gDW/h and ~4mmol/gDW/h, respectively, while in CHL evolved mutants are on average around 15mmol/gDW/h and 9mmol/gDW/h (pvalue= 0.0036). This is almost twice as in the wild type and certainly we observed an opposite trend in all the other populations evolved in glucose with or without antibiotics.

5. Fig. EV3: y-axis is odd. Plotted as %inhibition; 20% inhibited would grow faster than 40% inhibited, which I do not think is what the authors intended. Also, as it stands, this reviewer is not certain that statistical significance will be achieved with the data presented.

To investigate the potential role of fermentative metabolism in mediating tolerance to CHL we tested the inhibitory activity of the same concentration of CHL in cells growing in aerobic vs anaerobic batch. If fermentative metabolism was per se conferring tolerance to CHL we would expected to see lower growth inhibition of CHL under oxygen limitation. But this was not the case.

Instead, cells under oxygen limitation exhibited higher sensitivity (~20% vs 40% growth relative to the untreated conditions). The pvalue is now reported in the figure legend (Pvalue ttest = 0.0447). Even if barely significant, the point is that we can exclude the possibility that oxygen limitation (and hence fermentative metabolism) was per se a tolerance/resistance mechanism. Rather we reinforced our conclusion that this is an adaptive mechanism evolved in CHL resistant populations. We clarify figure axis and legend.

6. "Consistently, we showed that deletion of the efflux pump repressors MarR or AcrR (Nichols et al, 2011) not only rendered E. coli more tolerant to chloramphenicol (Appendix Fig. S7-9), but also caused a strong increase in glucose fermentation via acetate secretion (Fig. EV4)." This reviewer is concerned that the differences in S7 are not significantly different, and the differences in EV4 look like some could be significant but not others.

We agree with the reviewer that effect of *acrR* exhibited a mild (but significant) 2-fold increase in MIC, but the increases of ~80% and 40% are significant within the reported error (we report the p-value now in the legend). However, deletion of *acrA* or *acrB* reported in figure S8 are the two most detrimental gene deletion in the presence of Chloramphenicol, while *marR* deletion is the second most beneficial gene deletion. To address reviewers concerns we now performed a new series of experiment, in which we directly probe the level of expression and protein abundance of *acrAB* (see next point). We now replace figure S7 with a new figure.

7. *AcrR* and *MarR* deletion mutants are not substitutes for working with the *acrB* promoter mutation you found or *acrR* mutation. Data from the deletion mutants can be used in a supportive fashion, but not for drawing causal links about the populations you evolved. Without performing the experiments I mentioned previously with respect to whether *acrB* promoter mutation is necessary and sufficient (or mutation in *acrR*), causality is not established, and without causality little is learned about the antibiotic resistance that was evolved in your experiments.

We thank the reviewer for this comment. It is important to notice that our aim was not identification of specific beneficial mutations. Consciously our focus is on compensatory MECHANISMS, such as the identified rewiring of glucose metabolism in response to overexpression of the efflux pumps to achieve CHL resistance. While we understand that some researchers are more interested in the specific effects of single mutations, this is not our scope and it seems unfair to argue that causal mechanisms do not matter but only causal mutations. The reviewer has a point though that we did not establish for our evolved strains that their efflux pumps are indeed overexpressed. Hence, we performed two new experiments to verify this overexpression in (i) the evolved mutants and (ii) wildtype cells when perturbed with chloramphenicol, to provide further evidence for the MECHANISM causality.

Since multiple mutations in the region of the *acrAB* promoter alone or in combination can be responsible for increased expression of the *acrAB* efflux pumps, we directly measured the *acrB* protein abundance in all 12 strains evolved in glucose. Consistent with our expectations, *acrB* exhibited almost 4 times higher level of AcrB than in the wild-type strain, in contrast to ampicillin and norfloxacin resistant populations where the level of the protein remains unchanged (Fig 3). In addition, using a GFP reporter plasmid fused to the promoter region of *acrAB*, we demonstrated, that *acrAB* efflux pump plays a specific role in mediating the immediate response to chloramphenicol. In fact, *acrAB* is overexpressed upon chloramphenicol treatment, while it is not activated by norfloxacin.

Overall, we believe that the additional data support our claim that the specific rewiring of central carbon metabolism inferred from metabolomics data and experimentally validated in chloramphenicol resistant populations, works via the *acrAB* overexpression, which we now demonstrate by directly measuring protein abundance in the evolved resistant populations in glucose minimal medium. These experiments are not only of supportive fashion but actually crucial for our claim in a way that a mutation alone cannot be.

8. "Our EMC predictions (Fig. 4A) based on metabolite changes in resistant strains suggested recycling of anhydromuropeptides to play an important role in mediating resistance to ampicillin." Sensitivity to fosfomycin does not convincingly show that recycling of anhydromuropeptides is

important to mediating ampicillin resistance. Altering the level of anhydromuropeptides and then observing increases/decrease in ampicillin resistance would be convincing. Further, doing that in your evolved populations would establish causality.

We thank the reviewer for giving us a chance to clarify this point. It is not about the levels of anhydromuropeptides per se. But rather the flux capacity of the recycling pathway which can compensate for ampicillin resistance. Adding an enzyme inhibitor of the pathway is indeed a much more direct and informative way to test and support our predictions, than reviewer's suggestion of possibly supplementing anhydromuropeptides to wild-type or AMP resistant populations. Moreover, we now proved that the same hypersensitivity was not found for E.coli that has evolved AMP resistance in acetate minimal medium. In fact, for such strains we did not detect any changes in the recycling pathway.

9. " $\log_2(\text{OD}^* - (\text{OD}_{\text{fin}}/100))$ ": this is incorrect. Should not be a subtraction but a division to capture generations. I presume this is just a typo.

We thank the reviewer for spotting this mistake. We now corrected the typo.

Reviewer #3:

The present revision show some modest improvements without substantially addressing limitations in the data, analysis and figures. The paper does demonstrate that antibiotic resistance develops faster on glucose than acetate (an interesting result), and that there are antibiotic and carbon-source specific metabolic adaptations.

Correct. This has never been shown before.

As far as I can tell, it does not successfully validate recurrent antibiotic-resistance driving mutations,

It is worth noting the scope of our method was to hypothesize condition-dependent compensatory MECHANISMS (NOT mutations) for antibiotic resistance. Once more, we believe our work offers a complementary approach to study and understand evolution of antibiotic resistance with respect to a more classical "genetic approach".

nor does it clearly establish strong metabolic phenotypes associated with resistance to specific antibiotics, beyond a modest trend towards decreased respiratory metabolism with chloramphenicol resistance, which is pretty logical given that chloramphenicol blocks protein synthesis which is a major ATP consumer.

We found global metabolic rearrangements in antibiotic resistant populations that are clearly drug and media specific. This is shown and discussed already when investigating the pairwise similarity of metabolome profiles in evolved populations (Fig. 2 and EV1-2).

The distinct alteration of glucose respiration/fermentative metabolism observed in chloramphenicol resistant populations is not modest (see point 4 of reviewer 2), neither it is intuitive. The logic of the reviewer is unclear to us.

- First, the metabolic phenotype of an increased glucose uptake and acetate secretion is observed in the absence of chloramphenicol! This is certainly a result of genetic mutations not chloramphenicol per se. We now added new experimental evidences reinforcing the link between overexpression of efflux pumps *acrAB* and the switch from respiration to fermentative metabolism.
- Second, while it is correct that chloramphenicol blocks synthesis of proteins, and that this is an energetic costly process for the cells, it is certainly not so obvious that this would translate to higher glucose uptake and fermentation. According to the theory proposed in (Basan *et al*, 2015) this is likely not what is happening. Experimental evidences in (Basan *et al*, 2015) suggested that glucose fermentation is less costly in terms of proteome requirement. The effect of inhibition of protein synthesis is hence likely affecting the tradeoff between energy and protein costs. Chloramphenicol, by constraining the entire proteome, can induce a rearrangement of proteome resource allocation. As a result,

oxidative phosphorylation is reduced as it is less efficient (per protein) than fermentative metabolism. We already discuss different possibilities in the text.

The new method with shadow prices is more clear now, but I still do not find its output to be particularly persuasive. The authors do not validate any strengths of this new analysis approach.

We disagree with the reviewer, as two follow up investigations were performed to validate and investigate further the predictions made from our experimental/computational approach.

Finally, the authors continue to claim measurement of over 500 metabolites in the abstract, even though the text has now been corrected to reflect their actual methods.

We now write “putative metabolites” to avoid any overstatements

More importantly, they continue to rely on the ampicillin-Fos example in the abstract, even though they admit that the finding is weak.

We never admitted or said our finding was weak. But rather avoid to overstate that this result could be directly translated into clinic, which was never the goal of the present study. While the hypersensitivity of ampicillin resistant strains is between 6 and 2 fold higher than in wild-type (Fig. 4), this is a very robust result, which we have reinforced in the last revision, by showing that ampicillin resistance evolved in acetate minimal medium does not select for this hypersensitive phenotype.

We do agree with the reviewer that in the future more natural type of conditions should be investigated to assess whether such metabolic constraints on the evolution of antibiotic resistance can be translated in clinic (we explicitly commented on this). However, the existence of such type of constraints were never shown before, and could open new possibilities for the future design of new combinatorial treatments to fight resistant pathogens.

Overall, my impression is that this is a timely undertaking, but did not yet yield the type of substantive discoveries that one might have hoped for.

It is always difficult to anticipate what colleagues might hope for. Nevertheless, we would like to stress that by taking a rather original approach in this field of antibiotic resistance development, we made, in our view, quite some progress. The key results of our work are:

1. A new method (and its application to demonstrate relevance) to interpret metabolomics data via genome-scale models. This has never been done before and we envisage a broad application spectrum as a new tool for the metabolomics community.
2. Demonstration that metabolism constrains the evolution of antibiotic resistance - which has never been demonstrated before.
3. Applying our model-based method to the evolved populations, we managed to hypothesize condition-dependent compensatory MECHANISMS (NOT mutations) for antibiotic resistance. Two cases were validated by extensive independent experiments.

We can only hope that the reviewer agrees that these are novel achievements. Otherwise I would be helpful if the reviewer would be more specific on what type of results would have matched his “expectation”.

Thank you for submitting your revised study. We have now heard back from the two reviewers who were asked to evaluate your manuscript. The reviewers think that most major issues have been satisfactorily addressed. However, they still list some concerns, which we would ask you to address in a minor revision.

As you will see below, reviewer #2 still raises two major concerns (points #1 and #2). However, we think that your responses and analyses related to the link between *acrB* levels and chloramphenicol resistance and the recycling of anhydromuropeptides in ampicillin-glucose evolved strains, are satisfactory. In line with this, during our 'pre-decision cross-commenting' process, reviewer #1 mentioned that s/he thinks that the analyses proposed by reviewer #2 (points #1 and #2) seem outside the scope of a revision at this stage and emphasized that in his/her opinion the related issues have been satisfactorily addressed. As such, we do not think that these additional analyses requested by reviewer #2 are required for the acceptance of the study. We would nevertheless ask you to address the points raised by reviewer #1 and points #3 and #4 of reviewer #2.

REFEREE REPORTS

Reviewer #1:

The second revision provides further improvements and in my opinion addresses all major points raised by reviewers. Overall, I find the manuscript constitutes a significant contribution to antibiotic research, providing a new angle how metabolic constrains may be shaping antibiotic resistance development.

A few minor things that would be good to be addressed in final version in manuscript (and without the reviewers having to see implemented changes):

- i) diagonal on 2A/B cannot be set at 0, since this is the Pearson correlation between the same sample, which by definition is 1. I would personally show it as it should be - i.e 1; this is conventional practice in all correlation clusters I have seen. However, if authors prefer blanking this comparison, they can set it as NaN and a distinct color from heat map (grey?), but not to 0. This is very confusing and simply gives the wrong message.
- ii) please make sure you reread the new text and correct English and nomenclature. Some of the things I picked (there are more):
 - proteins cannot be deleted, only genes
 - "induces an overexpression" is not proper English
 - page 8; you may want to make the point clearer that Glu-CHL evolved populations have constitutively high expression of *AcrB* (so in absence of the drug) - and this is the difference between points i and iv.
 - first new sentence on page 9 needs rephrasing/breaking down in two. As is, it is incomprehensible.
- iii) Fig 3D: although effects are crystal clear, I would not calculate S.D. from 2 samples... You can just show the averages on the bar graph.
- iv) EV1: it would be good to explain what the whiskers represent; this is the most common variable in the box plot (the box itself is pretty standard).

Reviewer #2:

This reviewer appreciates the more thorough statistical analyses that were included in this second revision, and the measurement of *AcrB* abundance. As stated by the authors in the text, they sought to understand how metabolism influences the development of antibiotic resistance, and how mutations accumulated during development of resistance impact metabolism. The items I mention below are essential to drawing these connections.

1. As mentioned in my previous reviews, the authors have to work with the *acrB* promoter mutation(s) they observed in the evolved populations. The authors did show that the chloramphenicol evolved populations had higher acetate secretion and reduced respiration compared to controls, and they also showed that *AcrB* is in higher quantities in the chloramphenicol evolved populations. Those are good, but not enough to conclusively draw the connection between resistance

and metabolism. Other experiments are supportive, but not direct and they do not make a clear link between metabolism and resistance. The authors need to quantify chloramphenicol resistance, respiration, acetate secretion, and ideally AcrB abundance in the ancestral strain with the *acrB* promoter mutation moved in and evolved strains with the *acrB* promoter mutation fixed. This is the only way to connect metabolism and the resistance observed together. Further, since the authors have done these experiments with different strains, they are able to do them; these are not experiments outside the authors expertise.

2. The section describing the ampicillin and fosfomycin results is phenotypic. The fosfomycin results are interesting, but there is no clear connection between metabolism and resistance. Do the authors see increased flux through MurA? If so, what genetic mutation(s) would account for that? If they fix the mutation, does it reduce the flux and reduce ampicillin resistance?

3. For Figure 1C and associated EV1 the authors attempt to compare rates of evolution resistance by plotting the MIC as an exponential function of generations. Other than the norfloxacin data, it does not appear that an exponential function would be a good fit, and the rationale as to why it would be an exponential function eludes this reviewer. If the exponential function produces a poor fit, it would be rather unsettling to use such a function to compare the rates of evolution for statistical comparisons. Why don't the authors look at how many generations it takes to pass several resistance thresholds, such as 10xMIC, 20xMIC, 40xMIC, etc. Resistance is quantified by how many fold a strain's MIC is over a reference standard, and an analysis like this would be readily interpretable and more relevant in this reviewers mind than fitting the data to exponential curves, when the data does not appear to be exponential.

4. (Minor) For Fig S7, why was only the one low concentration of norfloxacin used, when a wide range of chloramphenicol concentrations are used?

3rd Revision - authors' response

28 January 2017

Reviewer #1:

The second revision provides further improvements and in my opinion addresses all major points raised by reviewers. Overall, I find the manuscript constitutes a significant contribution to antibiotic research, providing a new angle how metabolic constrains may be shaping antibiotic resistance development.

A few minor things that would be good to be addressed in final version in manuscript (and without the reviewers having to see implemented changes):

i) diagonal on 2A/B cannot be set at 0, since this is the Pearson correlation between the same sample, which by definition is 1. I would personally show it as it should be - i.e 1; this is conventional practice in all correlation clusters I have seen. However, if authors prefer blanking this comparison, they can set it as NaN and a distinct color from heat map (grey?), but not to 0. This is very confusing and simply gives the wrong message.

We have now use a distinct grey color as suggested by the reviewer.

ii) please make sure you reread the new text and correct English and nomenclature. Some of the things I picked (there are more):

- proteins cannot be deleted, only genes
- "induces an overexpression" is not proper English
- page 8; you may want to make the point clearer that Glu-CHL evolved populations have constitutively high expression of AcrB (so in absence of the drug) - and this is the difference between points i and iv.
- first new sentence on page 9 needs rephrasing/breaking down in two. As is, it is incomprehensible.

We revised the text and implemented suggested changes

iii) Fig 3D: although effects are crystal clear, I would not calculate S.D. from 2 samples... You can just show the averages on the bar graph.

Done

iv) EV1: it would be good to explain what the whiskers represent; this is the most common variable in the box plot (the box itself is pretty standard).

Done

Reviewer #2:

This reviewer appreciates the more thorough statistical analyses that were included in this second revision, and the measurement of AcrB abundance. As stated by the authors in the text, they sought to understand how metabolism influences the development of antibiotic resistance, and how mutations accumulated during development of resistance impact metabolism. The items I mention below are essential to drawing these connections.

1. As mentioned in my previous reviews, the authors have to work with the *acrB* promoter mutation(s) they observed in the evolved populations. The authors did show that the chloramphenicol evolved populations had higher acetate secretion and reduced respiration compared to controls, and they also showed that AcrB is in higher quantities in the chloramphenicol evolved populations. Those are good, but not enough to conclusively draw the connection between resistance and metabolism. Other experiments are supportive, but not direct and they do not make a clear link between metabolism and resistance. The authors need to quantify chloramphenicol resistance, respiration, acetate secretion, and ideally AcrB abundance in the ancestral strain with the *acrB* promoter mutation moved in and evolved strains with the *acrB* promoter mutation fixed. This is the only way to connect metabolism and the resistance observed together. Further, since the authors have done these experiments with different strains, they are able to do them; these are not experiments outside the authors expertise.

2. The section describing the ampicillin and fosfomycin results is phenotypic. The fosfomycin results are interesting, but there is no clear connection between metabolism and resistance. Do the authors see increased flux through MurA? If so, what genetic mutation(s) would account for that? If they fix the mutation, does it reduce the flux and reduce ampicillin resistance?

3. For Figure 1C and associated EV1 the authors attempt to compare rates of evolution resistance by plotting the MIC as an exponential function of generations. Other than the norfloxacin data, it does not appear that an exponential function would be a good fit, and the rationale as to why it would be an exponential function eludes this reviewer. If the exponential function produces a poor fit, it would be rather unsettling to use such a function to compare the rates of evolution for statistical comparisons. Why don't the authors look at how many generations it takes to pass several resistance thresholds, such as 10xMIC, 20xMIC, 40xMIC, etc. Resistance is quantified by how many fold a strain's MIC is over a reference standard, and an analysis like this would be readily interpretable and more relevant in this reviewers mind than fitting the data to exponential curves, when the data does not appear to be exponential.

Our strategy is more solid and general than the one proposed by the reviewer. It is important to notice that plots in Fig 1 are in log scale, and clearly exhibit a linear trend. That means that in absolute scale, increase resistance over generation time resembles an exponential curve. To clarify the doubts of this reviewer here below we report the estimated R^2 values from the exponential fitting of each biological replicate.

R ²					
NOR		AMP		CHL	
ACE	GLC	ACE	GLC	ACE	GLC
0.97	0.92	0.11	0.77	0.66	0.87
0.95	0.92	0.52	0.72	0.74	0.67
0.92	0.94	0.08	0.9	0.71	0.77
0.96	0.88	0.08	0.83	0.51	0.77

With the only exception of strains evolved in ampicillin acetate we obtained good fitting quality (here estimated by R^2). The low R^2 obtained in acetate ampicillin simply reflect the very little gain in resistance observed during the evolutionary experiment, and is here reflected (figure EV1) by estimates of ϕ close to 0. In other words a straight flat line would have been sufficient to describe the resistance trend in these populations.

4. (Minor) For Fig S7, why was only the one low concentration of norfloxacin used, when a wide range of chloramphenicol concentrations are used?

In order to show that induction of *acrB* transcription was specific to chloramphenicol we selected a dosage of norfloxacin close to MIC. This in order to compare *acrB* transcriptional induction to a scenario in which we expected the largest induction of the gene transcription by norfloxacin. Our results are consistent with phenotypic data reported in Nichols et al.

Thank you for sending us your revised manuscript. We are now satisfied with the modifications made and I am pleased to inform you that your paper has been accepted for publication.

Corresponding Author Name: Uwe Sauer

Manuscript Number: MSB-16-7028